# Unveiling Ecological and Genetic Novelty within Lytic and Lysogenic Viral Communities of Hot Spring Phototrophic Microbial Mats

Sergio Guajardo-Leiva,[a]* Fernando Santos,[b] Oscar Salgado,[a,c] Christophe Regeard,[d] Laurent Quillet,[e] Beatriz Díez[a,f,g]

[a]Department of Molecular Genetics and Microbiology, Pontificia Universidad Católica de Chile, Santiago, Chile
[b]Department of Physiology, Genetics, and Microbiology, University of Alicante, Alicante, Spain
[c]Laboratorio de Bioinformática, Facultad de Educación, Universidad Adventista de Chile, Chillán, Chile
[d]Université Paris-Saclay, CEA, CNRS, Institute for Integrative Biology of the Cell (I2BC), Gif-sur-Yvette, France
[e]Laboratoire Glyco-MEV EA 4358, Université de Rouen, Mont Saint Aignan, France
[f]Center for Climate and Resilience Research (CR)2, Santiago, Chile
[g]Center for Genome Regulation (CGR), Santiago, Chile

**ABSTRACT** Viruses exert diverse ecosystem impacts by controlling their host community through lytic predator-prey dynamics. However, the mechanisms by which lysogenic viruses influence their host-microbial community are less clear. In hot springs, lysogeny is considered an active lifestyle, yet it has not been systematically studied in all habitats, with phototrophic microbial mats (PMMs) being particularly not studied. We carried out viral metagenomics following *in situ* mitomycin C induction experiments in PMMs from Porcelana hot spring (Northern Patagonia, Chile). The compositional changes of viral communities at two different sites were analyzed at the genomic and gene levels. Furthermore, the presence of integrated prophage sequences in environmental metagenome-assembled genomes from published Porcelana PMM metagenomes was analyzed. Our results suggest that virus-specific replicative cycles (lytic and lysogenic) were associated with specific host taxa with different metabolic capacities. One of the most abundant lytic viral groups corresponded to cyanophages, which would infect the cyanobacteria *Fischerella*, the most active and dominant primary producer in thermophilic PMMs. Likewise, lysogenic viruses were related exclusively to chemoheterotrophic bacteria from the phyla *Proteobacteria*, *Firmicutes*, and *Actinobacteria*. These temperate viruses possess accessory genes to sense or control stress-related processes in their hosts, such as sporulation and biofilm formation. Taken together, these observations suggest a nexus between the ecological role of the host (metabolism) and the type of viral lifestyle in thermophilic PMMs. This has direct implications in viral ecology, where the lysogenic-lytic switch is determined by nutrient abundance and microbial density but also by the metabolism type that prevails in the host community.

**IMPORTANCE** Hot springs harbor microbial communities dominated by a limited variety of microorganisms and, as such, have become a model for studying community ecology and understanding how biotic and abiotic interactions shape their structure. Viruses in hot springs are shown to be ubiquitous, numerous, and active components of these communities. However, lytic and lysogenic viral communities of thermophilic phototrophic microbial mats (PMMs) remain largely unexplored. In this work, we use the power of viral metagenomics to reveal changes in the viral community following a mitomycin C induction experiment in PMMs. The importance of our research is that it will improve our understanding of viral lifestyles in PMMs via exploring the differences in the composition of natural and induced viral communities at the genome and gene levels. This novel information will contribute to

Address correspondence to Beatriz Díez, bdiez@bio.puc.cl.

*Present address: Sergio Guajardo-Leiva, Center for Bioinformatics and Integrative Biology, Life Sciences Faculty, Universidad Andres Bello, Santiago, Chile.

deciphering which biotic and abiotic factors may control the transitions between lytic and lysogenic cycles in these extreme environments.

**KEYWORDS** CRISPR, hot springs, lysogenic, lytic, phototrophic microbial mats, viral ecogenomics

Prokaryotes and their associated viruses are present in every environment on Earth, significantly altering the biosphere through their role in biogeochemical and nutrient cycles (1, 2). Viruses contribute to the fitness and evolution of their host via their infective processes, where the specific virus lifestyle has different effects on the host's ecology (1, 3, 4). A lytic or productive infection is characterized by an immediate replication and expression of the viral genome within the host cell, subsequently releasing new viral particles by cell lysis (5). In contrast, a lysogenic infection is characterized by integrating the viral genome into the host chromosome or as an episomal element, forming a new biological entity known as lysogen (5–7). This cycle does not produce viral particles immediately after infection; however, it can switch to a productive cycle depending on multiple factors, such as virus or host genetics, virus-host ratio, host physiological state, host density (quorum sensing), and environmental conditions (5–8).

Lytic viruses directly influence microbial community composition through predator-prey dynamics, in which the dominant or active taxa in the microbial community are selectively lysed, as described in the "kill-the-winner" ecological model (2, 6, 7, 9–11). Conversely, it has been proposed that in ecosystems with high nutrients and high microbial abundances, viruses will lysogenize the most metabolically active and sometimes dominant taxa in the microbial community following the "piggyback-the-winner" ecological model (6, 7). Lytic and lysogenic dynamics have been studied mainly in aquatic environments (marine and freshwater), while a few studies have focused on sediments and soils (1, 7, 8, 12, 13). The most used method to study lysogeny is to quantify viral production after temperate virus induction by mitomycin C (MitC), a DNA-damaging agent that triggers the SOS system (1, 7, 14). However, the molecular basis of mitomycin induction is known only for cultivable bacteria, and mitomycin sensitivity varies in a strain-specific manner (1, 7, 14). Therefore, the experimental study of lysogeny in natural viral communities remains challenging, and its difficulty directly increases with the diversity and complexity of the studied microbial community.

Hot spring environments harbor microbial communities dominated by a limited variety of microorganisms. Thus, these simplified systems have been used as models to study community ecology and to understand how biotic and abiotic interactions shape the microbial community structure (15, 16). In these environments, viruses are ubiquitous, numerous, and active microbial community components; therefore, hot springs have also become a model system to study viral populations (17–28).

Circumneutral hot springs within the thermophilic range ($\leq$70°C) are dominated mainly by phototrophic microbial mats (PMMs) (16, 23, 29–32). These biofilms are a consortium of different microbial groups vertically stratified and embedded in an organic matrix at the interface between the water and a solid substrate (16, 33). Oxygenic phototrophic cyanobacteria (i.e., *Synechococcus*, *Oscillatoria*, and *Fischerella*) and filamentous anoxygenic phototrophs (i.e., *Roseiflexus* and *Chloroflexus*) form the uppermost layer of the mat (16, 29, 34–36). A plethora of chemoheterotrophic bacteria and archaea form the deeper layers of the mat and subsequently interact with the primary producers through element and energy cycling (16). Viruses predating this biofilm type have been studied mostly through their host associations (21, 26, 37, 38). The only microbial mat viral metagenome that is currently publicly available revealed the presence of viruses that infect dominant primary producers, such as *Synechococcus*, *Roseiflexus*, and *Chloroflexus* (37). This agrees with what has been reported in PMMs from Porcelana hot spring (Northern Patagonia, Chile), where cyanophages make up one of the most abundant and transcriptionally active groups of the viral community (21). A recent survey using cellular metagenomics in the Manikaran hot spring

(Himachal Pradesh, India) shows that lysogeny is a dominant viral lifestyle in hot spring PMMs (26). However, due to the low coverage of these PMM metagenomes, a low diversity of viruses, which included 14 viral genomes from the *Myoviridae* and *Siphoviridae* families, was found (26).

To gain new insights into the identity and role of the primarily unexplored lytic and lysogenic viral communities of thermophilic PMMs, we conducted viral metagenomics on natural Porcelana hot spring PMMs at two different sites, as well as followed *in situ* MitC induction experiments performed in the same PMMs. Additionally, we examined the presence of integrated prophage sequences in 34 high-quality metagenome-assembled genomes (MAGs) recovered from Porcelana hot spring using published cellular metagenomes (21, 29, 31). Our analyses revealed genetic and compositional dissimilarities between viral communities at different PMM sites in Porcelana and between natural and experimentally induced viral communities. Altogether, our results showed that the lytic lifestyle was predominant in viral groups infecting primary producers of PMMs, which were also the most active and abundant phylum (*Cyanobacteria*) in the microbial mats. In contrast, the lysogenic lifestyle was predominant within viral groups infecting chemoheterotrophic bacteria, represented here by the phyla *Proteobacteria*, *Firmicutes*, and *Actinobacteria*. Finally, our results demonstrated that in this high-nutrient and high-microbial-abundance environment, both infection dynamics coexisted and were associated with the metabolism and consequently the ecological function of the host.

## RESULTS

**Taxonomic differences between natural and induced viral communities in Porcelana.** To understand and analyze the lifestyles of viral communities in Porcelana hot spring, we sequenced two sets of viral metagenomes obtained from natural and MitC-induced PMMs at 50°C and 55°C. These viral metagenomes represent one of the few viral data sets obtained so far in hot springs. A summary of the obtained sequences is presented in Table S1.

First, we performed a coverage analysis using Nonpareil software to evaluate if differential library coverage could affect the accuracy of comparative abundance analyses of the four viral metagenomes. The results showed that library coverages were in the range of 0.86 to 0.92, which was above the recommended coverage threshold of 0.6. Furthermore, the coverage differences between data sets were less than 2-fold, confirming that it was possible to accurately detect differentially abundant sequences between these metagenomes. Second, we estimated the degree of genetic differentiation between both sets of samples using the MinHash distances of k-mer counts. Hierarchical clustering analyses using those genetic distances showed two separate clusters grouped by the site (P50 and P55) regardless of MitC induction (Fig. 1A). Third, reads were taxonomically classified against the NCBI Nucleotide database. Despite the low number of mapped reads (in the range of 155,000 to 937,000), most were classified as *Caudovirales* (Fig. 1B). Precisely, most of the mapped sequences from the natural communities at the P50 and P55 sites aligned with viral proteins of the *Podoviridae* family (77% and 60%, respectively). However, this family was less represented in the MitC-induced communities, accounting for only 42% and 14% of the reads at P50 and P55, respectively (Fig. 1B). Conversely, in both MitC-induced communities, the *Myoviridae* and *Siphoviridae* families were more represented (Fig. 1B). Additionally, sequences from the single-stranded DNA (ssDNA) groups represented by circular replication-associated protein-encoding single-stranded (CRESS) viruses and the *Microviridae* family were present only in the P55 site and with a higher representation in the MitC-induced community.

**Differences between natural and induced viral community structure revealed by protein clusters and viral operational taxonomic units.** We constructed a set of protein clusters (PCs) and viral operational taxonomic units (vOTUs) to analyze and quantify the differences between the viral protein universe, viral species, and population structure across the natural and MitC-induced viral communities, including known

 

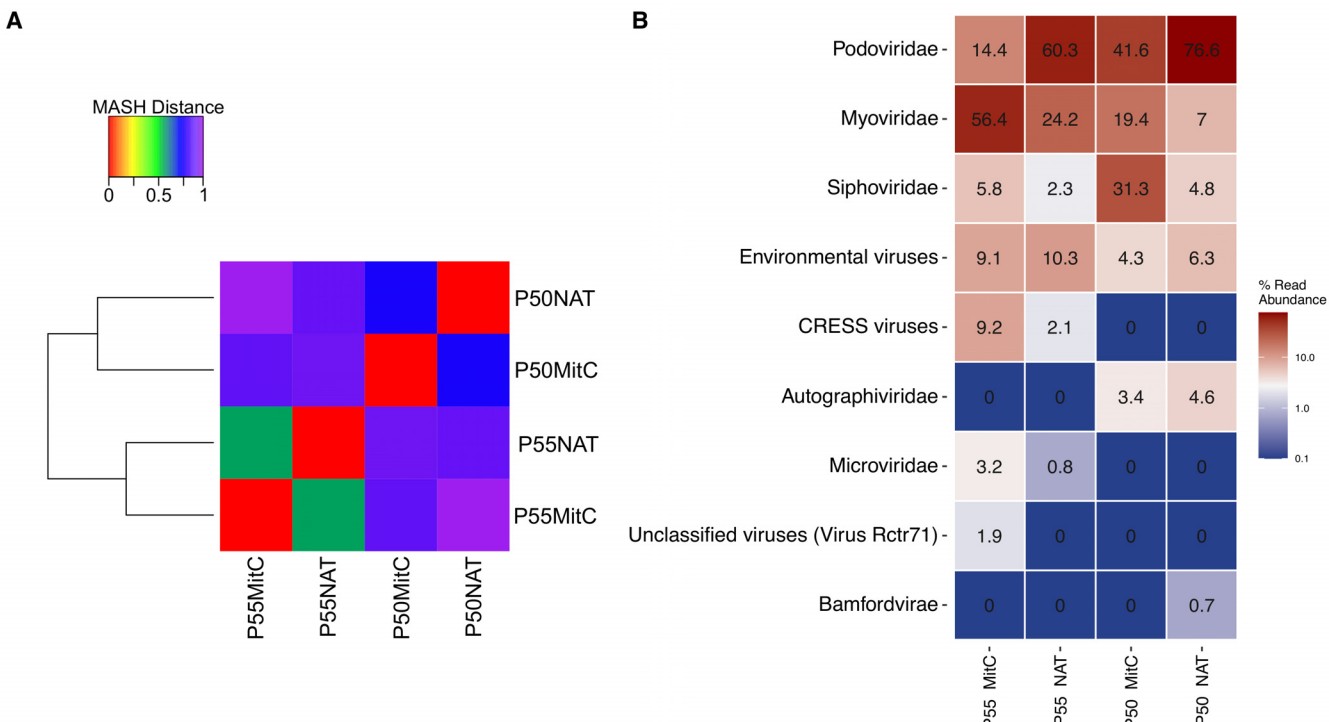

**FIG 1** Genetic and taxonomic dissimilarities of Porcelana hot spring viral communities, measured through k-mer frequencies and alignment to NCBI Nucleotide database. (A) Hierarchical clustering of mitomycin C induced and natural samples from two sites. The dendrogram was constructed based on a matrix of MASH distances from metagenomic reads using Ward's minimum variance method. Heatmap colors represent the pairwise MASH distance. (B) Relative abundances of viral families in Porcelana viral metagenomes from natural and mitomycin C induced communities. Metagenomic reads were taxonomically classified by the LCA algorithm through local alignment to NCBI Nucleotide database. Heatmap colors represent the relative abundance in a logarithmic scale. Abundances were normalized by library size.

and unknown sequences. A total of 9,505 PCs of two or more proteins were obtained, representing 26% to 52% of the quality-filtered metagenomic reads (Table S1). The largest PC contained 15 proteins, while 66% of the PCs were formed by only two proteins. Functional assignment of the PCs via the Pfam database allowed us to annotate 1,296 clusters to 615 protein family models (Table S2). The most abundant clusters corresponded to DNA metabolism proteins (e.g., integrases, terminases, helicases, and DNA polymerases) and viral structural proteins (e.g., tail proteins, portal proteins, and capsid proteins). The vOTU data set comprised 755 species-level consensus viral genomes, representing 25% to 44% of the total quality-filtered reads from the different samples (Table S1).

PCs and vOTUs were used to quantify and compare Shannon's diversity, Pielou's evenness, and species/functional richness between samples. Despite not being statistically significant, functional (PC) diversity and evenness increased in the MitC-induced communities relative to those in the natural communities at both sites (Fig. S1). However, richness decreased at P50 and increased at P55 in induced communities compared to that in natural ones. For the vOTUs, the diversity, evenness, and richness decreased in the MitC-induced community at the P50 site compared with those in the natural community. Conversely, at the P55 site, all these viral species metrics increased for the MitC-induced community.

Functional (PC) and species (vOTU) compositional dissimilarities were evident between the different sites (P50 and P55) and also between the different conditions (natural and MitC-induced), as determined by principal coordinates analysis (PCoA) using Bray-Curtis distances (Fig. 2A and B). The first axis separated the communities according to sites (P50 and P55), explaining 59.9% and 55.5% of the variance for the PCs and vOTUs. The second axis separated samples according to the natural or induced conditions, explaining 29.3% of the total variance for PCs and vOTUs. Additionally,

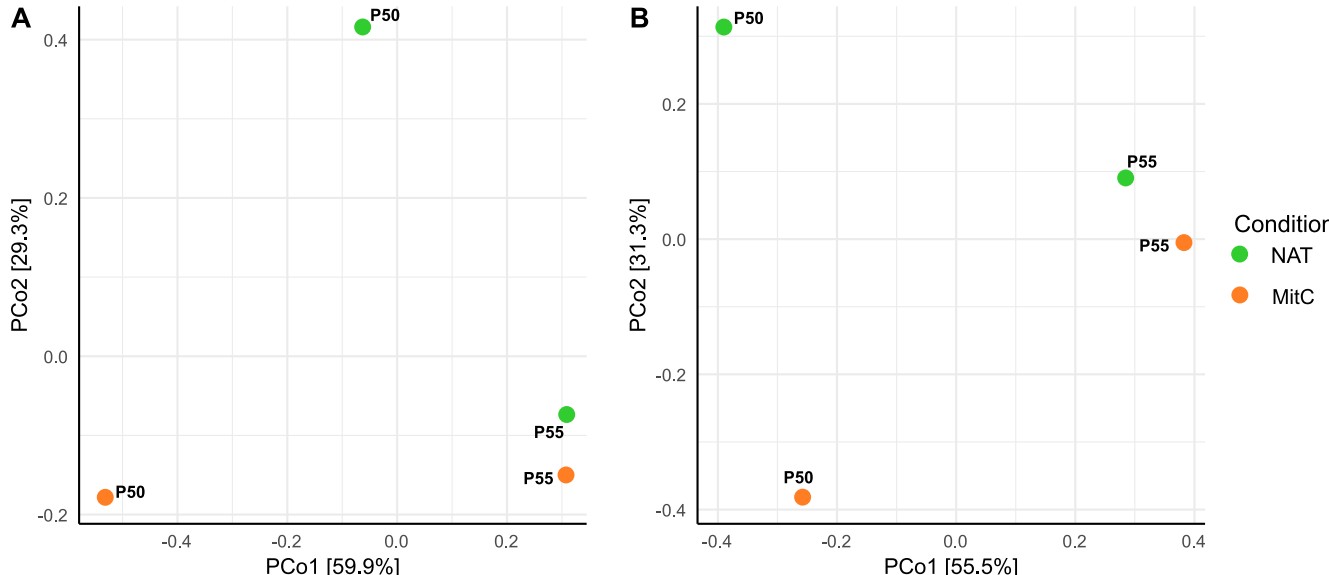

**FIG 2** Compositional dissimilarities of Porcelana hot spring viral communities, measured through PCs and vOTUs based on to Bray-Curtis dissimilarity matrices. (A) Principal coordinate analysis of mitomycin C induction and natural samples from two sites based on PCs Bray-Curtis distance matrix. (B) Principal coordinate analysis of mitomycin C induced and natural samples from two sites based on vOTUs Bray-Curtis distance matrix. For both PCoAs, no initial data transformation has been applied. The relative contribution (eigenvalue) of each axis to the total inertia in the data is indicated in percent at the axis titles.

none of the vectors associated with environmental variables (pH/temperature) fit with the ordination of the viral communities ($P > 0.05$, 9,999 permutations). A rank-based permutation test (BDM test) of the PC and vOTU abundance distribution corroborates previous results (Table S3), with a statistically significant difference ($P < 0.001$) in the composition of the PCs and vOTUs between different sites and conditions. A *post hoc* test (Wilcoxon test with Bonferroni correction) determined which pairwise differences were statistically significant ($P < 0.001$), where PC distribution was significantly different between all samples, including any site and condition combinations (Table S4). Instead, vOTU distribution was significantly different between the natural viral communities and the MitC-induced communities at both sites (Table S4). Taken together, these results show that sampling sites contributed to the variance in viral community structure to a greater extent (55 to 60%) than MitC induction, which accounted for about 30% of the variance.

**Identification of lysogens at genome and gene levels.** As compositional differences were found between natural and induced communities in Porcelana PMMs, the next objective was to assess whether these differences were due to changes in identifiable lysogenic molecular markers. For this purpose, specific markers, such as integrases, recombinases, and ParA/B genes, in the PC data set and prophage sequences in the vOTU data set were searched using Venn diagrams and differential abundance analysis (Fig. 3, Fig. S2 and S3, and Table S5). Venn diagram analyses showed that ~13% of the total PCs and ~21% of the total viral species were shared between all viral communities (Fig. 3). Likewise, communities from the same sampling site shared a more substantial number of PCs (up to 37%) and vOTUs (up to 38%) than samples under the same condition (natural or MitC-induced; up to 25% for PCs and 28% for vOTUs). A total of 1,224 PCs were exclusive to the induced communities, from which 68 were annotated (Table S5), but none of them corresponded to any known lysogenic marker (e.g., integration or recombination proteins). Likewise, 180 vOTUs were found exclusively in the induced communities, but only two were identified as prophages (scored as intact) by PHASTER analysis (Table S5).

To corroborate these results, we searched for lysogenic markers and prophage genomes in the whole PC and vOTU data sets. PCs annotated as lysogenic markers

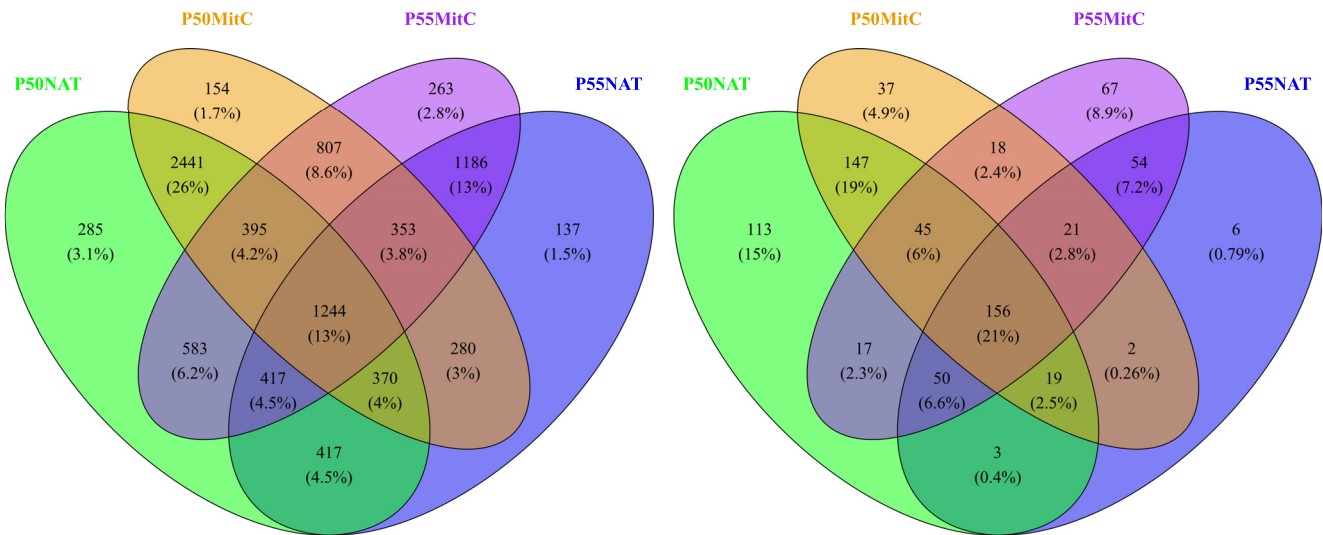

**FIG 3** Venn diagrams of the total PCs (A) and vOTUs (B) from natural and mitomycin C induced communities. Percentages between parentheses represent the contribution of the different conjuncts of PCs or vOTUs to the total number of PCs and vOTUs.

were shared between natural and induced conditions and did not show an association between abundances and the induction condition (Fig. S2). Only transposase genes showed a higher abundance in MitC-treated samples, especially at site P50. An analysis of singleton proteins (Table S6) revealed the same trend. Only one Mu-like phage proteins and one integrase were found at higher abundance in the induced samples. Likewise, 12 new vOTUs were classified as complete prophages by PHASTER (Table 1 and Fig. S3), in which 10 of these putative prophages showed no relationship between their abundance and the induction condition.

**TABLE 1** Summary results of the viral gene network analysis for the putative temperate viruses (PHASTER and differential analyses) in Porcelana hot spring[a]

| vOTU | Contig | Identification method | Closest genome | Cluster host | Cluster family |
|------|--------|----------------------|----------------|--------------|----------------|
| vO1 | #_P55_C_19 | PHASTER | *Bacillus* phage phi105c | *Firmicutes* | *Siphoviridae* |
| vO24 | *_P55MIT_C_7 | Phyloseq to edgeR | *Bacillus* phage BCD7 | *Firmicutes* | *Myoviridae/Siphoviridae* |
| vO25 | *_P50MIT_C_112 | Phyloseq to edgeR | *Bacillus* virus G | *Firmicutes* | *Myoviridae/Siphoviridae* |
| vO26 | #_P50MIT_C_17 | PHASTER | *Brevibacillus* phage Osiris | *Firmicutes* | *Myoviridae* |
| vO27 | *_P50MIT_C_92 | Phyloseq to edgeR | *Clostridium* phage c-st | *Firmicutes* | *Siphoviridae* |
| vO28 | *_P55MIT_C_60 | Phyloseq to edgeR | *Clostridium* phage c-st | *Firmicutes* | *Siphoviridae* |
| vO29 | *#__P55MIT_C_27 | Both | *Clostridium* phage PhiS63 | *Firmicutes* | *Siphoviridae* |
| vO30 | #_P50MIT_C_20 | PHASTER | *Lactobacillus* phage phiLdb | *Firmicutes* | *Siphoviridae* |
| vO31 | *#_P55MIT_C_34 | Both | *Erysipelothrix* phage SE-1 | *Firmicutes* | *Siphoviridae* |
| vO32 | *_P50MIT_C_133 | Phyloseq to edgeR | *Geobacillus* virus E3 | *Firmicutes* | *Siphoviridae* |
| vO33 | #_P50MIT_C_11 | PHASTER | *Lactobacillus* phage phiAQ113 | *Firmicutes* | *Myoviridae* |
| vO34 | #_P50MIT_C_16 | PHASTER | *Enterobacteria* phage 13a | *Proteobacteria* | *Podoviridae* |
| vO35 | #_P50_C_15 | PHASTER | *Pseudomonas* phage PAJU2 | *Proteobacteria* | *Siphoviridae* |
| vO36 | #_P50MIT_C_13 | PHASTER | *Pseudomonas* phage PaMx28 | *Proteobacteria* | *Siphoviridae* |
| vO37 | #_P50MIT_C_5 | PHASTER | *Pseudomonas* phage phiPSA1 | *Proteobacteria* | *Siphoviridae* |
| vO38 | #_P50_C_7 | PHASTER | *Pseudomonas* phage phiPto-bp6g | *Proteobacteria* | *Myoviridae* |
| vO39 | #_P50_C_77 | PHASTER | *Vibrio* phage K139 | *Proteobacteria* | *Myoviridae* |
| vO40 | #_Burkholderiaceae_GJ-E10 | PHASTER | P50MIT_C_171 | *Proteobacteria* | NA |
| vO41 | #_P55MIT_C_15 | PHASTER | *Burkholderia* virus BcepMu | *Proteobacteria* | *Myoviridae/Siphoviridae* |
| vO42 | #_P50_C_10 | PHASTER | *Rhodococcus* phage ReqiPoco6 | *Actinobacteria/Firmicutes* | *Siphoviridae* |
| vO43 | *_P50MIT_C_63 | Phyloseq to edgeR | P55MIT_C_56 | NA | NA |
| vO44 | *_P55MIT_C_56 | Phyloseq to edgeR | P50MIT_C_63 | NA | NA |
| vO45 | *_P55MIT_C_99 | Phyloseq to edgeR | Singleton | NA | NA |

[a]Table shows taxonomic classification of the cluster genomes and their putative host taxonomy. The closest viral genome was obtained from the node located at the shortest distance in the network. NA = not available.

Subsequently, we identified PCs and vOTUs with abundances in induced samples statistically significantly higher than those in natural samples (Fig. S2 and S3). Our analysis showed that 12 PCs and 10 vOTUs (*_vOTUs in Table 1) were statistically more abundant in MitC-induced communities. These PCs were mostly unknown or associated with nonlysogenic functions, such as terminase, peptidase, FtsK/SpoIIIE, and tubulin/FtsZ. Two of the 10 vOTUs (Table 1) were also found by PHASTER analysis; therefore, only these possess the complete genomic characteristics observed in prophage sequences from the databases.

Finally, we analyzed the potential presence of prophage sequences integrated into the bacterial genomes recovered from Porcelana hot spring PMMs using genome-resolved metagenomics techniques to identify MAGs from three published Porcelana cellular metagenomes (21, 29, 31). From those, we obtained 34 high-quality MAGs ($\geq$90% genome completeness and $\leq$10% contamination), which were distributed over nine phyla (Table 2). Only one complete prophage was detected using PHASTER (Table 1). This complete prophage was found in MAG 30 from the *Burkholderiaceae* family (genus GJ-E10) within the *Betaproteobacteria* class (Table 2). The prophage sequence (PP_Burkholderia_GJ-E10) was 28.2 Kb long and had 43 predicted proteins, including a transposase, an integrase, and other *Caudovirales* hallmark genes, including the tail, capsid, and portal proteins. Additionally, several other incomplete prophages (IPP) sequences were found in the other MAGs (Table 2), but most lacked identifiable integrases, recombination proteins, or integration sequences (tRNAs and attachment sites).

Overall, our analyses recovered 23 plausibly lysogenic viral genomes (complete and incomplete), representing 1 integrated into a host genome, 10 shared between natural and induced communities, 2 exclusives to the induced communities, and finally, 10 statistically more abundant in the induced communities than in the natural ones. This presence-abundance of lysogenic markers and putative prophages in natural and induced communities points to lysogeny as a common viral lifestyle in hot spring PMMs.

**Host and taxonomic assignment of Porcelana viral genomes through network analysis.** To cluster and visualize closely related viral genomes (vOTUs) from natural and MitC-induced Porcelana viral communities, we used 775 vOTUs and 2,304 reference genomes of archaeal and bacterial viruses to build a gene-sharing network implemented on vConTACT2 (39). The final network contained 2,698 nodes (455 vOTUs and 2,243 reference genomes), which formed 168 clusters that can be considered viral genera (Fig. 4). The remaining 320 vOTUs and 61 reference genomes represented singletons and were excluded from the network. Most Porcelana vOTUs grouped into 103 new viral genera (clusters), while only 86 formed clusters with the reference genomes and were subsequently considered known viral genera. Host information was deduced for the 34 Porcelana MAGs by mapping CRISPR spacers to the vOTU data sets and clustering vOTUs to the reference genomes in the network. These strategies allowed us to classify the 23 putative lysogenic vOTUs (Table 1) and the 23 most abundant (relative abundance of $\geq$1%) vOTUs in Porcelana (Table 3).

Lysogenic vOTUs were associated with clusters formed by *Firmicutes* (11) and *Proteobacteria* (eight) reference viruses, belonging mostly to the *Siphoviridae* and *Myoviridae* families (Table 1). Most vOTUs that were statistically more abundant in the MitC-induced communities were classified as part of the *Firmicutes* (seven) or unknown (three) clusters.

Analysis of the 23 most abundant vOTUs showed that 12 were classified to viral families inside the order *Caudovirales* (5 *Podoviridae*, 3 *Myoviridae*, and 4 *Siphoviridae*), while the other 11 remained unclassified (Table 3). Host information based on reference genomes or CRISPR spacers showed that the most abundant viruses in Porcelana were infecting *Actinobacteria* (three vOTUs, 8.5% abundance), *Cyanobacteria* (six vOTUs, 7.2% abundance), *Proteobacteria* (three vOTUs, 3.5% abundance), *Firmicutes* (one vOTU, 3.3% abundance), and *Chloroflexi* (one vOTU, 1% abundance). These results show that the most abundant vOTUs infect critical components of the PMMs, such as the cyanobacterium *Fischerella* and unknown members of the phylum *Actinobacteria*. Likewise, most vOTUs identified as lysogenic were associated with viral clusters that infect chemoorganoheterotrophic bacteria within the phyla *Firmicutes* and *Proteobacteria*.

**TABLE 2** Temperate virus search analyses of metagenome-assembled genomes (MAGs) from Porcelana hot springs[a]

| MAG | Phylum | Order | Family | Genus | Length (Mb) | MAG %GC | No. viral regions | Completeness | Relevant protein(s) | Length (Kb) | %GC | tRNA | Attachment site | No. total proteins | No. viral proteins |
|---|---|---|---|---|---|---|---|---|---|---|---|---|---|---|---|
| 1 | Acidobacteria | Solibacterales | Solibacteraceae | | 4.17 | 64.37 | 2 | Incomplete | Integrase | 14.9 | 65.7 | 0 | Yes | 20 | 10 |
| | | | | | | | | | Tail; virion | 9.9 | 68.8 | 0 | No | 10 | 9 |
| 2 | Armatimonadetes | Fimbriimonadales | GBS-DC | GBS-DC | 3.82 | 53.88 | 1 | Incomplete | Virion | 13.4 | 63.25 | 0 | No | 14 | 10 |
| 3 | Armatimonadetes | Fimbriimonadales | GBS-DC | GBS-DC | 2.46 | 61.75 | 1 | Incomplete | NA | 14.3 | 59.97 | 0 | No | 16 | 9 |
| 4 | Armatimonadetes | Fimbriimonadales | GBS-DC | UBA10441 | 2.66 | 60.93 | 1 | Incomplete | Portal; head; capsid | 7.1 | 54.13 | 0 | No | 10 | 7 |
| 5 | Bacteroidetes | Chitinophagales | Saprospiraceae | | 3.79 | 54.72 | 0 | | | | | | | | |
| 6 | Bacteroidetes | Chlorobiales | Chloroherpetonaceae | | 3.04 | 47.75 | 0 | | | | | | | | |
| 7 | Bacteroidetes | Cytophagales | Cyclobacteriaceae | UBA2336 | 3.48 | 47.33 | 1 | Incomplete | NA | 7.3 | 43.47 | 0 | No | 10 | 6 |
| 8 | Bacteroidetes | Kapabacteriales | NICIL-2 | NICIL-2 | 2.62 | 56.12 | 0 | | | | | | | | |
| 9 | Bacteroidetes | Cytophagales | Cyclobacteriaceae | UBA2336 | 3.15 | 46.37 | 1 | Incomplete | Lysin; cI-like repressor | 8.5 | 46.37 | 0 | No | 8 | 7 |
| 10 | Bacteroidetes | Chitinophagales | Saprospiraceae | UBA10441 | 3.65 | 54.79 | 0 | | | | | | | | |
| 11 | Chloroflexi | Anaerolineales | SBR1031 | A4b | 4.46 | 54.19 | 1 | Incomplete | Head | 6.3 | 54.61 | 0 | No | 8 | 6 |
| 12 | Chloroflexi | Thermoflexales | | | 4.35 | 63.79 | 1 | Incomplete | Tail | 9.8 | 63.22 | 0 | No | 7 | 6 |
| 13 | Chloroflexi | Thermoflexales | | | 4.30 | 63.79 | 2 | Incomplete | Tail | 9.8 | 63.22 | 0 | No | 7 | 6 |
| | | | | | | | | | NA | 11.2 | 65.53 | 0 | No | 9 | 6 |
| 14 | Chloroflexi | Thermoflexales | | | 5.30 | 64.68 | 4 | Incomplete | Tail | 9.6 | 65.3 | 0 | No | 7 | 6 |
| | | | | | | | | | NA | 10 | 62.29 | 0 | No | 11 | 8 |
| | | | | | | | | | Tail | 9 | 68.8 | 0 | No | 7 | 6 |
| | | | | | | | | | Transposase | 10.1 | 67.06 | 0 | No | 14 | 8 |
| 15 | Chloroflexi | Chloroflexales | Roseiflexaceae | Roseiflexus | 4.88 | 59.37 | 1 | Incomplete | NA | 9.6 | 58.14 | 1 | No | 9 | 6 |
| 16 | Chloroflexi | Chloroflexales | Roseiflexaceae | Roseiflexus | 4.86 | 59.44 | 2 | Incomplete | NA | 9.6 | 58.15 | 1 | No | 9 | 6 |
| | | | | | | | | | Capsid; terminase | 6.1 | 55.39 | 1 | No | 12 | 6 |
| 17 | Chloroflexi | Chloroflexales | Roseiflexaceae | Roseiflexus | 4.97 | 58.15 | 3 | Incomplete | NA | 9.6 | 58.15 | 0 | No | 9 | 6 |
| | | | | | | | | | Lysin; tail | 10.8 | 43.45 | 0 | No | 13 | 7 |
| | | | | | | | | | Tail; capsid; portal; terminase | 14.7 | 43.08 | 0 | No | 10 | 7 |
| 18 | Chloroflexi | Chloroflexales | Chloroflexaceae | Chloroflexus | 4.82 | 55.98 | 1 | Incomplete | NA | 5.5 | 54.25 | 0 | No | 6 | 6 |
| 19 | Chloroflexi | Chloroflexales | Chloroflexaceae | Chloroflexus | 4.83 | 55.85 | 1 | Incomplete | NA | 5.5 | 54.25 | 0 | No | 6 | 6 |
| 20 | Chloroflexi | Chloroflexales | Chloroflexaceae | Chloroflexus | 4.98 | 55.85 | 4 | Incomplete | Terminase; portal; head; capsid | 13.1 | 47.39 | 0 | No | 13 | 8 |
| | | | | | | | | | Integrase | 15.2 | 52.4 | 0 | Yes | 19 | 12 |
| | | | | | | | | | NA | 5.5 | 54.25 | 0 | No | 6 | 6 |
| | | | | | | | | | Capsid; head; portal | 6.2 | 54.15 | 0 | No | 10 | 7 |
| 21 | Cyanobacteria | Pseudophormidiales | Pseudophormidiaceae | | 6.02 | 53.53 | 1 | Incomplete | NA | 6 | 55.5 | 0 | No | 6 | 6 |
| 22 | Cyanobacteria | Cyanobacteriales | Nostocaceae | Fischerella | 5.49 | 40.97 | 1 | Incomplete | NA | 8.6 | 39.58 | 0 | No | 8 | 6 |
| 23 | Cyanobacteria | Cyanobacteriales | Nostocaceae | Fischerella | 5.26 | 41.1 | 4 | Incomplete | NA | 7.6 | 40.78 | 0 | No | 9 | 6 |
| | | | | | | | | | Integrase-resolvase; transposase | 11.1 | 38.62 | 0 | Yes | 8 | 6 |
| | | | | | | | | | NA | 8.3 | 39.17 | 0 | No | 8 | 6 |
| | | | | | | | | | NA | 5.5 | 47.12 | 0 | No | 6 | 6 |
| 24 | Cyanobacteria | Cyanobacteriales | Nostocaceae | Fischerella | 5.40 | 41.11 | 3 | Incomplete | Protease; Integrase; Transposase | 25.8 | 40.14 | 0 | Yes | 8 | 6 |
| | | | | | | | | | NA | 7.6 | 40.79 | 0 | No | 9 | 6 |
| | | | | | | | | | NA | 8.6 | 39.58 | 0 | No | 9 | 6 |
| 25 | Deinococcus-Thermus | Deinococcales | Thermaceae | Meiothermus | 3.89 | 59.25 | 3 | Incomplete | Terminase; portal; capsid; tail | 14.7 | 43.12 | 0 | No | 10 | 7 |
| | | | | | | | | | Tail; lysin | 10.8 | 43.51 | 0 | No | 12 | 7 |
| | | | | | | | | | Integrase | 21.4 | 62.98 | 0 | Yes | 7 | 6 |
| 26 | Planctomycetes | Isosphaerales | Isosphaeraceae | | 5.69 | 64.35 | 1 | Incomplete | Head; capsid; tail | 11.5 | 66.37 | 0 | No | 14 | 10 |
| 27 | Planctomycetes | Phycisphaerales | UBA11161 | | 3.49 | 52.45 | 1 | Incomplete | DprA; tail fiber | 8.3 | 50.87 | 0 | No | 8 | 6 |

**TABLE 2** (Continued)

| MAG | Phylum | Order | Family | Genus | Length (Mb) | MAG %GC | No. viral regions | Completeness | Relevant protein(s) | Length (Kb) | %GC | tRNA | Attachment site | No. total proteins | No. viral proteins |
|---|---|---|---|---|---|---|---|---|---|---|---|---|---|---|---|
| 28 | Planctomycetes | Phycisphaerales | UBA1161 | | 3.30 | 52.45 | 1 | Incomplete | Tail | 8.5 | 50.06 | 0 | No | 8 | 6 |
| 29 | Planctomycetes | Phycisphaerales | UBA1161 | | 3.44 | 52.76 | 3 | Incomplete | Tail | 8.5 | 50.06 | 0 | No | 8 | 6 |
| | | | | | | | | Incomplete | DprA; tail fiber | 8.3 | 50.87 | 0 | No | 8 | 6 |
| | | | | | | | | Incomplete | Tail fiber | 7.6 | 72.17 | 0 | No | 9 | 7 |
| 30 | Proteobacteria | Betaproteobacteriales | Burkholderiaceae | GJ-E10 | 2.97 | 68.57 | 1 | Intact | Virion; tail; capsid; portal; transposase; integrase | 28.2 | 68.61 | 0 | Yes | 43 | 26 |
| 31 | Proteobacteria | Acetobacterales | Acetobacteraceae | Elioraea | 4.01 | 71.41 | 1 | Incomplete | Capsid; tail | 18.5 | 71.71 | 0 | No | 24 | 18 |
| 32 | Proteobacteria | Acetobacterales | Acetobacteraceae | Elioraea | 4.02 | 72.59 | 3 | Incomplete | Lysin; terminase; cl-like repressor | 7.7 | 74.09 | 0 | No | 11 | 6 |
| | | | | | | | | | Transposase | 4.5 | 68.68 | 0 | No | 9 | 6 |
| | | | | | | | | | NA | 6.7 | 69.44 | 0 | No | 9 | 7 |
| 33 | Proteobacteria | Geminicoccales | Geminicoccaceae | | 3.87 | 72.68 | 2 | Incomplete | Head; capsid; tail | 9.9 | 71.24 | 0 | No | 11 | 7 |
| | | | | | | | | | NA | 12.2 | 69.68 | 0 | No | 17 | 10 |
| 34 | Verrucomicrobia | Pedosphaerales | UBA9464 | | 4.20 | 67.14 | 1 | Incomplete | Transposase | 6 | 66.07 | 0 | No | 9 | 6 |

aTable shows taxonomic classification, GC content, and length of each MAG. Temperate virus regions were classified as intact or incomplete based on PHASTER score, and relevant characteristic of temperate viruses regions such as length, GC content, presence of tRNAs, and attachment sites appear in the table. NA, not available.

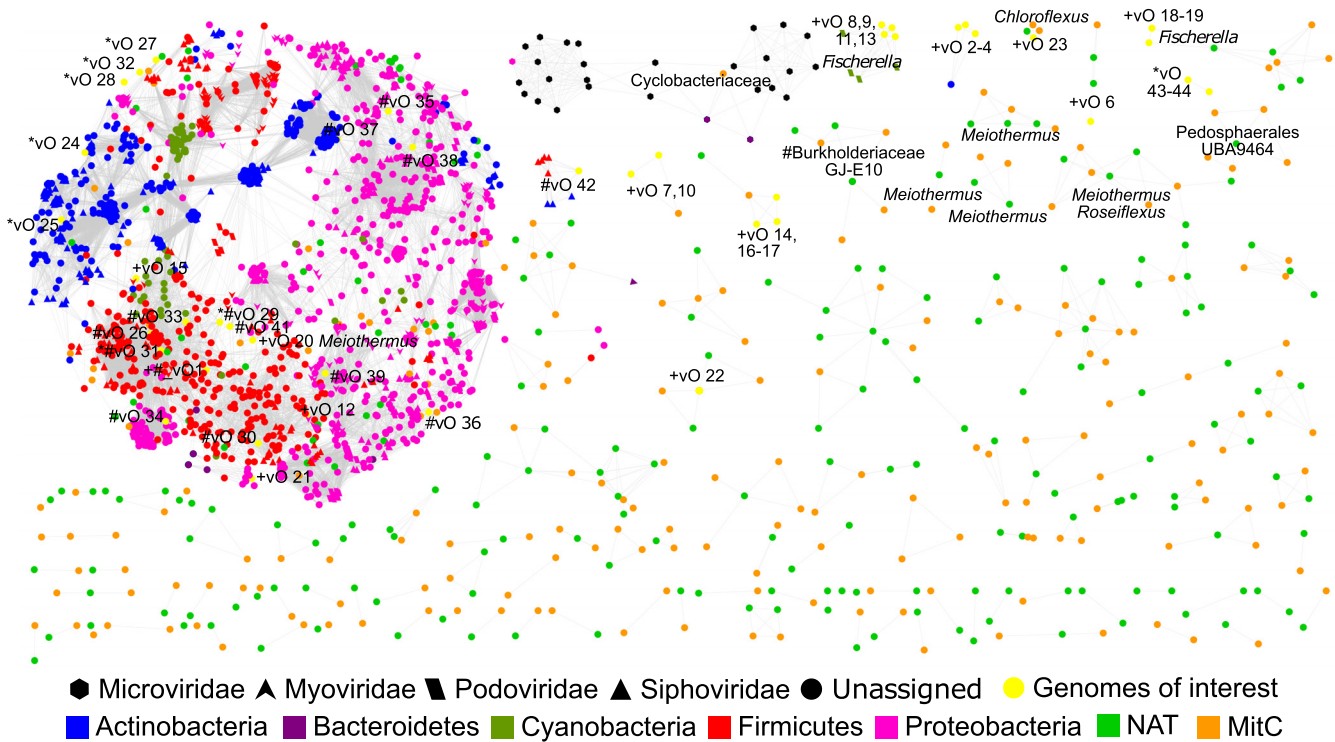

**FIG 4** Protein-sharing network of Porcelana vOTUs. Each node represents a vOTU or reference genome. Reference genomes were classified by viral family and host information (node form and colors), while host names in black letters were obtained by mapping CRISPR spacers of Porcelana MAGs to vOTUs. Edges between nodes indicate a statistically significant relationship between the protein profiles of the viral genomes. Modules within the network are composed of groups of similar sequences using the ClusterOne algorithm. vOTUs with relative abundances above 1% are marked by a plus symbol (+), vOTUs identified by the differential abundance analyses are marked by an asterisk symbol (*), vOTUs identified by PHASTER are marked by a hash symbol (#), and vOTUs identified by both analyses are marked by an asterisk followed by a hash symbol (*#).

**Host-control and viral recombination/integration systems revealed by genomic annotation of abundant and lysogenic vOTUs.** To better understand the gene structures and functions carried by the most abundant vOTUs (Fig. 5 and Table S6) and the lysogenic vOTUs (Fig. S4 and Table S7 [https://github.com/phageattack/Porcelana-viromes]) from Porcelana, we annotated these genomes and constructed schematic gene maps. For vOTUs belonging to a genetic cluster, only the longest genome of the cluster was plotted.

For the most abundant vOTUs in Porcelana (Fig. 5 and Table S6), three of these genomes showed terminal redundancy, specifically from *Actinobacteria* (vO4) and *Cyanobacteria* (vO18 and vO19), and carried several recognizable hallmark genes as well as some genes related to replication, recombination, nucleotide metabolism, lysis, and DNA modification. Other vOTUs related to *Proteobacteria*-associated viruses (vO12, vO15, and vO20) belonged to three distinct, unrelated clusters, showing hallmark genes and integration-related functions, such as transposase and regulatory Mu-like genes. A *Firmicutes*-associated vOTU (vO1) presented hallmark genes and functions associated with lysis and four genes annotated as transposases. Finally, a vOTU associated with *Chloroflexi* (vO23) showed hallmark genes, as well as replication, lysis, DNA modification, nucleotide metabolism, host recognition, integration, and accessory genes (FtsK/SpoIIIE).

Among the 23 lysogenic vOTUs (Fig. S4 and Table S7 [https://github.com/phageattack/Porcelana-viromes]), two displayed terminal redundancy (vO33 and vO34). Viral OTUs that were recognized exclusively by differential abundance analysis generally lacked genes related to integration and recombination systems. In some cases, it was only possible to annotate nonstructural genes related to replication, lysis, or accessory functions. This last characteristic was directly related to the length of the vOTU (e.g., vO25, vO27, vO28, vO43, vO44, and vO45). Among those associated with *Firmicutes* (vO1 and vO24 to vO33), a

**TABLE 3** Summary results of the viral gene network analysis of the most abundant (≥1%) vOTUs in Porcelana hot spring[a]

| Contig | vOTU | P50NAT (counts) | P50MitC (counts) | P55NAT (counts) | P55MitC (counts) | Total abundance (%) | CRISPR | Closest genome | Reference lifestyle | Host (reference or CRISPR) | Reference family |
|---|---|---|---|---|---|---|---|---|---|---|---|
| P55_C_19 | vO1 | 16 | 2,973 | 3,299 | 527 | 3.3 | NA | Bacillus phage phi105c | Lysogenic | Firmicutes | Siphoviridae |
| P50MIT_C_30 | vO15 | 0 | 2,488 | 0 | 0 | 1.2 | NA | Rhodobacter phage RcapMu | Lysogenic/transposable | Proteobacteria | Siphoviridae |
| P50_C_313 | vO20 | 87 | 47 | 1,098 | 838 | 1.0 | Meiothermus | Rhodovulum phage vB_RhkS_P1 | Lysogenic | Thermus/Proteobacteria[b] | Siphoviridae |
| P50MIT_C_6 | vO2 | 52 | 6,086 | 0 | 8 | 3.0 | NA | Tetrasphaera phage TJE1 | Lytic | Actinobacteria | Myoviridae |
| P55MIT_C_21 | vO3 | 61 | 5,932 | 0 | 26 | 2.9 | NA | Tetrasphaera phage TJE1 | Lytic | Actinobacteria | Myoviridae |
| P50_C_11 | vO4 | 51 | 5,606 | 0 | 7 | 2.8 | NA | Tetrasphaera phage TJE1 | Lytic | Actinobacteria | Myoviridae |
| P50_C_26 | vO8 | 1,061 | 575 | 888 | 244 | 1.4 | Fischerella | Anabaena phage A-4L | Lytic | Cyanobacteria | Podoviride |
| P55MIT_C_12 | vO9 | 868 | 500 | 1,076 | 307 | 1.3 | Fischerella | Anabaena phage A-4L | Lytic | Cyanobacteria | Podoviride |
| P55_C_10 | vO11 | 820 | 500 | 1,058 | 280 | 1.3 | Fischerella | Anabaena phage A-4L | Lytic | Cyanobacteria | Podoviride |
| P55_C_79 | vO12 | 2,386 | 181 | 35 | 52 | 1.3 | NA | Thalassomonas phage BA3 | Lytic | Proteobacteria | Podoviridae |
| P50MIT_C_23 | vO13 | 927 | 541 | 915 | 241 | 1.3 | Fischerella | Anabaena phage A-4L | Lytic | Cyanobacteria | Podoviridae |
| P50_C_225 | vO5 | 11 | 64 | 1,728 | 2,755 | 2.2 | NA | Singleton | NA | NA | NA |
| P50_C_52 | vO6 | 2,508 | 450 | 0 | 0 | 1.4 | NA | P55_C_102 | NA | NA | NA |
| P55_C_104 | vO7 | 1,827 | 924 | 106 | 96 | 1.4 | NA | P55MIT_C_163 | NA | NA | NA |
| P55MIT_C_163 | vO10 | 1,634 | 897 | 93 | 105 | 1.3 | NA | P55_C_104 | NA | NA | NA |
| P55MIT_C_71 | vO14 | 168 | 264 | 1,459 | 723 | 1.3 | NA | P55_C_43 | NA | NA | NA |
| P55_C_43 | vO16 | 154 | 230 | 1,358 | 692 | 1.2 | NA | P55MIT_C_71 | NA | NA | NA |
| P50MIT_C_69 | vO17 | 146 | 240 | 1,340 | 690 | 1.2 | NA | P55_C_43 | NA | NA | NA |
| P50_C_42 | vO18 | 1,062 | 902 | 49 | 124 | 1.0 | Fischerella | P55MIT_C_22 | NA | Cyanobacteria | NA |
| P55MIT_C_22 | vO19 | 1,062 | 899 | 47 | 124 | 1.0 | Fischerella | P50_C_42 | NA | Cyanobacteria | NA |
| P55_C_100 | vO21 | 1,856 | 91 | 27 | 35 | 1.0 | NA | Rhodoferax phage P26218 | Lytic | Proteobacteria | Siphoviridae |
| P50MIT_C_37 | vO22 | 11 | 0 | 1,965 | 0 | 1.0 | NA | P50_C_239 | NA | NA | NA |
| P55_C_1 | vO23 | 21 | 55 | 95 | 1,800 | 1.0 | Chloroflexus | P55MIT_C_3 | NA | Chloroflexi | NA |

[a]Table shows relative abundance of vOTUs in each sample and percentage of relative abundance in the total vOTUs set. Taxonomic classification of the host, based on the MAGs taxonomy, is provided for vOTUs that have a CRISPR spacer hit. Closest viral genome was obtained from the node located at the shortest distance in the network. NA, not available.
[b]CRISPR spacers found in Meiothermus sp. MAG Bacillus phage SPBc2.

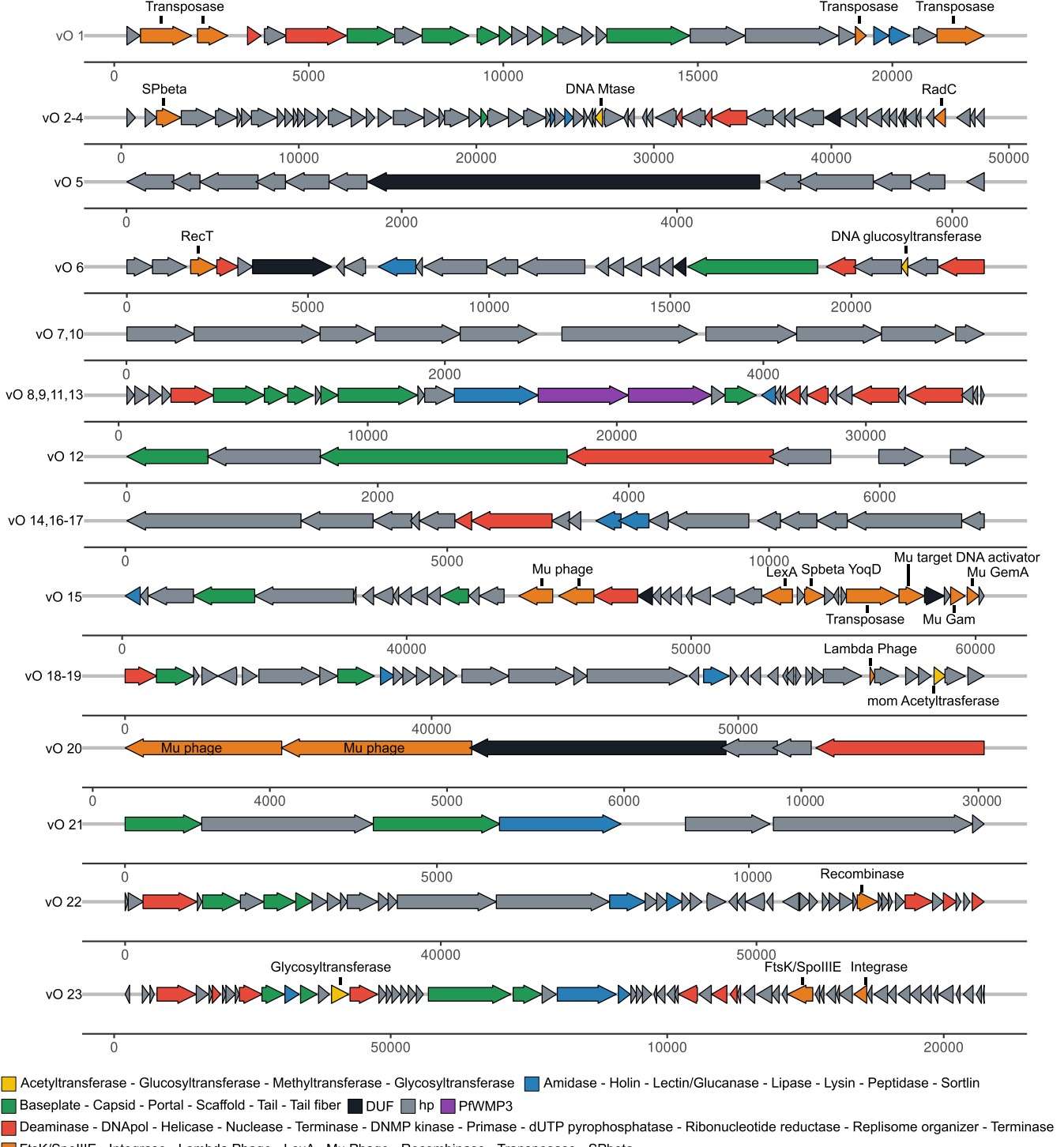

**FIG 5** Gene map of the most abundant vOTUs across all samples. Only vOTUs with relative abundances of ≥1% were considered. Arrows represent ORFs and color the functional annotation. ORFs associated with functions or proteins from lysogenic viruses appear in orange.

recombination/integration system was identified in four, two of which (vO1 and vO26) also presented genes annotated as transposases, while the other two (vO29 and vO30) presented genes annotated as recombinases. Interestingly, these vOTUs contained accessory genes to influence sporulation (FtsK/SpoIIIE, vO24 and vO29) and biofilm formation (SinR, vO26), processes that were associated with their hosts. *Proteobacteria*-associated viruses (vO34 to vO41) corresponded only with contigs detected by PHASTER and presented

genes encoding recombinases as integration/recombination systems with three genomes carrying transposase genes (vO36, vO40, and vO41). Instead, the *Actinobacteria*/*Firmicutes*-associated virus (vO42) did not have a recognizable integration/recombination system. Finally, two unidentified genomes (vO43 and vO44) presented genes encoding recombinases, and interestingly, they also contained genes (FtsK/SpoIIIE) that could influence the sporulation process of their host.

## DISCUSSION

Despite the viromics revolution that has been ongoing for the last decade, viral communities from thermophilic PMMs have remained unexplored. Furthermore, combining field experiments and viral metagenomics to link the replicative cycles of these viral communities with their taxonomy and putative hosts has never been undertaken.

In other environments, lysogeny has been estimated by quantifying the viral progeny after prophage excision using the DNA-damaging agent MitC (1, 6, 12–14, 40) and references therein. However, the effects of this methodology in natural communities are not well understood (5, 9). Therefore, the identification and quantification of lysogens in mixed natural communities have been challenging. To circumvent this, here, we used viral metagenomic analyses to explore and unveil the genetic and compositional dissimilarities between natural and MitC-induced viral communities from PMMs at two different sites (50°C and 55°C). Together, our results suggest that the lytic lifestyle was dominant in viral populations that infect the most active and abundant primary producers in these mats, such as *Cyanobacteria*, while the lysogenic lifestyle was common among PMM viral populations associated with chemoheterotrophic members of the phyla *Proteobacteria*, *Firmicutes*, and *Actinobacteria*.

**Genetic, taxonomic, and compositional dissimilarities of natural and induced viral communities.** Analyses based on k-mer frequencies and taxonomic classification of viral metagenomic reads (Fig. 1) revealed significant genetic distances (k-mer frequencies) between hot spring sites but not between the natural and MitC-induced communities. Conversely, the most pronounced taxonomic differences were found between natural and MitC-induced conditions.

These genetic and taxonomic differences in viral communities within Porcelana PMM sites were previously reported and explained by changes in the host community structure (21). Likewise, a similar pattern, in which host taxonomic dissimilarities at different sites generate differences in archaeal viruses, has been reported in acidic hot springs of Yellowstone National Park (18, 41) and the Manikaran hot spring in the Himalayas (26).

In Porcelana, one of the most dramatic taxonomic differences between the natural and induced communities was a lower representation of viral reads associated with the *Podoviridae* family in the MitC-induced samples. These differences translate explicitly into a reduced lytic cyanophage abundance, described previously as infecting the cyanobacteria genus *Fischerella* in this hot spring (21). These *Podoviridae* cyanophages are part of a cosmopolitan group in this type of hot spring and have also been reported in Brandvlei hot spring, South Africa (27, 28). Furthermore, to avoid bias due to the low number of sequences with taxonomic annotations, we cataloged viral sequences using a database-independent approach by quantifying PCs and vOTUs. The results unveiled stronger compositional dissimilarities between sites than between conditions (Fig. 2). Statistical tests (BDM and Wilcoxon) confirmed the significance of the global dissimilarities found in our exploratory multivariate analyses. Changes in host composition can explain these compositional differences between the viral communities at different sites according to the environmental gradients across the hot spring (21, 29). Since the initial natural microbial communities are different between sites, it is expected that the induced viral communities will also differ. These differences can increase due to the dissimilar MitC effects on diverse taxa (5, 42) or even within strains of the same species (7). The latter was observed in the dissimilar changes in alpha diversity after mitomycin treatment for the different Porcelana sites. Thus, in communities with high sensitivity to mitomycin, a decrease in diversity and its components is generated, as a few induced viruses are highly abundant. On the contrary,

there is an increase in diversity and its components in communities with low sensitivity to mitomycin because induced viruses coexist with lithic viruses, having more similar abundances.

It is important to consider that PC- and vOTU-based metrics represent different components in the viral communities. PCs, which use smaller contigs, can capture functions codified by the rare virosphere. In contrast, vOTUs usually represent the most abundant viral genotypes, as large contigs in metagenomic assemblies are expected to be formed mainly by the abundant viral populations (43). Consequently, the robustness of the observations made here is more remarkable since it considers the differences generated by both the most abundant viral components and the rare ones.

These results emphasize that, under the current experimental setup, MitC induction and other potential factors that were not quantified in this study affected the viral community structure at both sites but to a lesser extent than the biotic and abiotic conditions inherent to the P50 and P55 sites. However, exploring how these differences would translate into changes at the gene and genome levels for the different viral groups is necessary.

**Lysogeny is a common viral lifestyle in hot springs PMMs.** Lysogeny has been suggested as a common feature in hot spring microbial communities (19, 20, 26, 44). However, lysogeny has been experimentally demonstrated only for planktonic microbial communities from hot springs (74°C to 82°C) in California (20) and PMMs in the current work. *In situ* MitC inductions in California hot springs showed up to a 1.4-fold increase in number of viral-like particles (VLPs) by epifluorescence microscopy, demonstrating the existence of lysogens susceptible to MitC in these environments. Additionally, temperate viruses have been found more frequently in indirect studies by searching lysogenic markers (e.g., integrases and recombinases) (25, 26, 44). Very recently, the combination of single-cell genomics and cellular metagenomics in low-mobility environments with high microbial abundances, such as the nonphototropic thermophilic microbial mat of Little Hot Creek (California), has shown that lysogeny is the predominant viral lifestyle (19).

In Porcelana, lysogenic markers were found in both the natural communities (with a high abundance of integrases, ParB, and recombinases) and MitC-induced communities (where transposases were consistently more abundant) (Fig. S2). Furthermore, we detected statistically significant differences in proteins from MitC inductions compared to those from natural communities, such as the accessory genes (FtsK/SpoIIIE and tubulin/FtsZ) that were more abundant in the induced communities than in the natural ones. Interestingly, both genes have been described previously in lysogenic but also in lytic viruses. The accessory gene FtsK/SpoIIIE is highly conserved between *Bacillus* phages (45) and is involved in the lytic cycle suppression and phage genome segregation into the highly resistant spore (46, 47). Likewise, the tubulin/FtsZ gene is associated with the partitioning of nonintegrated prophage genomes during *Clostridium* cell division (48) and with relocating replicating phage chromosomes to the center of the cell to form an efficient phage factory that improves the burst size (49).

Similarly, some of the vOTUs identified as lysogenic by differential abundance analysis were also found using PHASTER and associated with lysogenic reference genomes despite not presenting molecular markers of this lifestyle. The latter implies that vOTUs with differential abundances possibly correspond to incomplete genomes that escape traditional detection methods based on molecular markers of lysogeny, indicating that these statistical methods (differential abundance) are biologically meaningful. However, the discovery of some lysogenic vOTUs in Porcelana with higher relative abundances in natural communities than in the induced ones indicates that only a fraction of the viruses with lysogenic characteristics were sensible to MitC induction and also the existence of natural induction factors. This phenomenon can be explained in terms of the MitC dosage, which can vary on a species- and strain-specific basis, leading to unsuccessful induction or inhibition of the host and viral production (7, 50). Likewise, the common induction of some groups of temperate phages in response to

unknown environmental factors is also plausible and in line with the high abundance of natural lysogens reported in other hot springs (19, 26).

Moreover, MAG analysis revealed a low presence of integrated viruses, in which only one prophage was recovered from the 34 Porcelana MAGs despite their high quality. The lack of genome completeness cannot explain this low frequency of integrated prophages since the MAGs recovered here were of better quality than those recovered with other commonly used thresholds (14). Nevertheless, we found many nonactive or vestigial prophage sequences among the Porcelana MAGs. These vestigial sequences, which usually lack structural proteins (exempting tail-related proteins) and lysogeny markers (e.g., integration enzymes, recombination enzymes, tRNAs, and attachment sites) (51), can still provide adaptive functions to bacteria, such as gene transfer agents, type 6 secretion systems, and bacteriocins (52). Another possible explanation for this low number of prophage sequences found in Porcelana MAGs is the inherent difficulty in obtaining mobile elements in MAGs generated from short-read sequencing (53).

**Unexplored and ecologically relevant viral genera in Porcelana.** A monopartite protein shared network helped us explore and visualize species connections, taxonomical clustering, and host-virus pairs in Porcelana. Through this analysis, we grouped different Porcelana viral genomes (vOTUs) into viral clusters that were considered to represent the same genus with 80% precision (54), allowing us to discover 103 new viral genera using taxonomic relationships to reference genomes (RefSeq-ICTV viral genomes). This novelty rate is even higher than that recently reported for the marine environment (55), even though most of the new genera discovered in this study will probably remain utterly unknown due to the lack of any taxonomic or host-related information.

Nonetheless, the integration of abundance analyses, genomic annotation, and vConTACT network allowed us to identify the taxonomic connections, hosts, and integration and host-control strategies of the 23 most abundant viral species (≥1%) in Porcelana PMMs at 50°C and 55°C. The most abundant viral genus (vO2 to 4) was associated with *Actinobacteria*, based on clustering information to the reference genome from *Tetrasphaera TJE1* lytic phage (56). Members of *Actinobacteria* are part of the rare biosphere of Porcelana (29), yet their viruses have previously been reported as an abundant and active group in this hot spring (21). Although the reference genome that these vOTUs resemble is lytic, it is necessary to consider that its abundance was 10-fold higher in the induced community than in the natural community, supporting further studies related to this type of infectious cycle. Likewise, the presence of the RadC gene in the genome annotation of these viruses (vO2 to 4) points to the possibility of new and unknown mechanisms of integration of these viral genomes into the bacterial chromosome (57). The second most abundant group of viruses in Porcelana was represented by two genera of lytic phages infecting the cyanobacterium *Fischerella* (vO8, 9, 11, 13, 18, and 19). As in many other hot springs worldwide, *Fischerella* plays an essential function in Porcelana PMMs as a primary producer (carbon and nitrogen fixers) and community builder (29, 30, 58). One of these two cyanophage genera (vO8, 9, 11, and 13) had a representative genome within the *Podoviridae* family (TC-CHP58, GenBank: KY888885) recovered previously from Porcelana cellular metagenomes (21).

Furthermore, *Proteobacteria* viruses (vO12, 15, 20, and 21) represented the third most abundant group recovered and were close to viruses from different species of purple non-sulfur bacteria. Specifically, two of these viral species (vO15 and vO20) were closely affiliated with Mu-like phages, transposable lysogenic viruses (59, 60), which was corroborated by the multiple Mu-like proteins annotated in their genomes. Terminase genes of Mu-like phages have also been reported in other hot springs, such as the Brandvlei hot spring (28), but the vOTU recovered here represents the first thermophilic Mu-like genome available to date. Firmicutes viruses, represented by one vOTU (vO1), were also abundant in Porcelana and were closely affiliated with a thermophilic *Geobacillus* phage (61). Interestingly, this vOTU genome harbored transposases

and was the only recognizable lysogenic phage (based on PHASTER results) with an abundance of ≥1% in Porcelana. Finally, it is worth mentioning that the vOTU representing a *Chloroflexus* phage (vO23), which has no known homologs in the databases, showed an integrase and the accessory gene FtsK/SpoIIIE in its genome annotation, which is typical of *Firmicutes* lysogenic phages (45). So far, the only virus associated with *Chloroflexus* corresponds to prophage sequences from genomes of *C. aggregans* DSM 9485 and *C. aurantiacus* J10fl, but it is unknown whether or not they are active (62). Considering that the relative abundance of this vOTU increased almost 19-fold in the induced community at site P55, it may be the first functional *Chloroflexus* prophage ever discovered.

It is important to note that the three genomes within the group of abundant vOTUs associated with lysogenic lifestyles (as identified through network analysis and genomic annotation) are transposable-like viruses. Likewise, the cyanophage genomes described here did not show any evidence of following a lysogenic cycle, either by the annotation of lysogeny markers in their genomes or by differential abundance in the induced samples. The latter was reinforced by the absence of prophages in the Porcelana *Cyanobacteria* MAGs and other thermophilic *Cyanobacteria* genomes available at NCBI (http://phast.wishartlab.com/Download.html).

Finally, there were eight (vO5, 6, 7, 10, 14, 16, 17, and 22) of the 23 most abundant vOTUs for which it was not possible to establish a lifestyle based on references. Two of these eight vOTUs (vO6 and vO22) showed the presence of recombinases, which could suggest a lysogenic lifestyle.

Our results agree with previous studies from hot springs and other environments, pointing to *Proteobacteria* and *Firmicutes* as the most common hosts of lysogenic viruses in these environmental communities. These two phyla harbor the vast majority of prophage sequences in the databases (51), oceans (63), and hot springs (26) so far. Like here, Sharma and collaborators found that 28 of the 31 lysogenic viral genomes found in the metagenomes from Manikaran hot spring microbial mats and sediments were associated with *Proteobacteria* and *Firmicutes* (26). However, temperate phage richness in Porcelana seems to be lower than that in the Manikaran hot springs, possibly due to the low activity of *Proteobacteria* and *Firmicutes* in Porcelana (29).

Altogether, our data suggest the coexistence in these hot spring PMMs of both lytic infection dynamics that follow the kill-the-winner model and lysogenic dynamics that follow the piggyback-the-winner model. We consider that although the limitations of our study do not allow us to be conclusive about these dynamic processes, they do allow us to elaborate a theoretical discussion of lytic-lysogenic dynamics in hot springs PMMs.

In PMMs, nutrients are not a limiting factor for chemoheterotrophic bacteria, being an environment where bacterial abundance and growth rates are higher than in oligotrophic ones. Consequently, piggyback-the-winner dynamics predict that lysogeny will be favored in these nutrient-rich environments. However, cyanobacteria, being the primary producers in these mats, face several constraints to their growth, such as the solubility of gases and chlorophyll degradation at high temperatures. Due to these constraints, hot spring environments are highly competitive for cyanobacteria generating a high microdiversity in these microorganisms (31). Therefore, the high lytic activity in cyanobacterial layers is also predicted to increase diversification through antagonistic evolutionary mechanisms by kill-the-winner dynamics contributing to the microdiversity found in these microorganisms in hot springs. Thus, the metabolism of each host would be related to the lifestyle of its viruses, where viruses of primary producers (e.g., *Cyanobacteria*) would follow kill-the-winner dynamics. In contrast, viruses of chemoheterotrophic bacteria (e.g., *Proteobacteria*, *Firmicutes*) would follow piggyback-the-winner dynamics.

We acknowledge the limitations of the present study, which are mainly summarized in the lack of biological replicates that account for the heterogeneity of PMMs and the lack of adequate controls in terms of the conditions under which mitomycin C

treatment was performed (bottle effect). A sufficient number of biological replicates and controls would have allowed us to separate the effect of Mitomycin treatment from other effects associated with the experimental design. However, the results presented here provide valuable information on lytic and potentially lysogenic viral communities prevalent in model biological systems such as PMMs. Thus, our results shed light on the influence of biotic factors on the structure of these model microbial communities, providing specific information at the level of viral genes and genomes. Finally, our study leaves at the disposal of the scientific community hundreds of viral genomes that can be used in further studies to deepen the role of viruses in both microbial mat studies of thermal systems and biofilms in general.

**Conclusions.** The present work represents the first analysis of viral metagenomes from *in situ* viral induction experiments in a hot spring that couples genome-resolved metagenomics with viral metagenomes to understand viral cycles in PMMs. Here, we show that the second most abundant viral group has a lytic lifestyle and follows the kill-the-winner ecological model. These viruses infect *Fischerella* spp., some of the most active and abundant PMM microorganisms, which physically support the mat and are responsible for nitrogen-carbon fixation. Moreover, we corroborate that in hot springs, as in many other environments, lysogeny is a broadly distributed state within viral populations that infect chemoheterotrophic bacteria of the phyla *Proteobacteria* and *Firmicutes*. These phyla take advantage of the abundant nutrients derived from primary producers in these thermophilic PMMs; therefore, their associated lysogenic viruses better fit the piggyback-the-winner model.

## MATERIALS AND METHODS

**Sample collection, viral enrichment, and *in situ* mitomycin C induction.** Porcelana hot spring (42° 27′29.1′′S, 72°27′39.3′′ W), located in the Chilean Patagonia, had a circumneutral pH range of 6.8 to 7.1 and temperatures ranging from 46°C to 60°C when sampled in December 2014. Viral communities were sampled from PMMs growing in ponds where the surface water reached 50°C (pH 7.24, site P50) and 55°C (pH 6.67, site P55), as described in reference 21. Briefly, 5 L of interstitial fluid (the liquid trapped inside the mats) was obtained by squeezing the microbial mat with all its layers through a 150-$\mu$m sterilized polyester net (SEFAR PET 1000; Sefar, Heiden, Switzerland). The interstitial fluid was sequentially filtered through 0.8- and 0.22-$\mu$m pore size polycarbonate filters (Millipore, Milford, MA, USA). Particles in the 0.22-$\mu$m filtrate were concentrated to a final volume of 35 mL using a tangential-flow filtration cartridge (Vivaflow 200, 100-kDa pore size; Vivascience, Lincoln, UK).

MitC experiments were performed *in situ* at the hot spring. Briefly, 500 cm$^3$ of the PMMs (with all its layers) from P50 and P55 (separately) were allowed to enter freely inside 2-L transparent polycarbonate bottles containing site-specific hot spring water to a total volume of 1 L to allow gas exchange. Microbial mats were separated only from their anchoring point to avoid as much as possible the disruption of the mat structure. MitC was added to each bottle at a final concentration of 1 $\mu$g/mL. The bore of each bottle was covered with a 10-$\mu$m nylon net, and the bottles were incubated for 24 h *in situ* at their respective pond sites with the bottleneck above the water surface of the hot spring. After incubation, the microbial mat and water from each bottle were squeezed and filtered as described above.

Due to the high concentrations of exopolysaccharides in interstitial fluids, which generate sample autofluorescence and aggregation of viral-like particles (VLPs), it was impossible to perform VLP counts by epifluorescence microscopy.

**Purification of viral particles, DNA extraction, and sequencing.** Viral particles were purified by CsCl density gradient ultracentrifugation as described in reference 64. Briefly, samples were centrifuged at 60,000 × *g* for 2 h at 4°C in a swinging bucket rotor (SW 40 Ti; Beckman Coulter, Indianapolis, IN, USA). The viral CsCl fraction was treated with 300 U of DNase I for each 1 ml of sample. Viral DNA was then extracted using the formamide/cetyltrimethylammonium bromide (CTAB) method, followed by the phenol-chloroform method (64). The quality and quantity of the DNA were checked by agarose electrophoresis, spectrophotometric absorbance (Nanodrop), and fluorometric (Qubit) assays. The DNA was then stored at −80°C for future use. Bacterial DNA contamination was checked by 16S rRNA gene PCR amplification using a universal bacterial primer set (515F: 5′-GTGYCAGCMGCCGCGGTAA-3′; 806R: 5′-GACTACNVGGGTWTCTAAT-3′) (https://earthmicrobiome.org/protocols-and-standards/16s/).

Bacterial DNA (*Escherichia coli* JM109) was used as a PCR spike control to check for PCR inhibition by inhibitors that may have coprecipitated with the viral DNA.

Purified viral DNA was sequenced using Illumina MiSeq technology at the Roy J. Carver Biotechnology Center (IL, USA). Briefly, shotgun viral DNA libraries were prepared with the KAPA Hyper Prep kit (Kapa Biosystems, Wilmington, MA, USA). Libraries were pooled, quantified by quantitative PCR (qPCR), and sequenced on one MiSeq flowcell for 251 cycles from each end of the fragments.

Cutadapt (65) was used for quality filtering, leaving only sequences longer than 50 bp (-m 50), as well as for 3′ end trimming of bases with a quality below 30 (-q 30) and hard clipping of the first nine leftmost bases (-u 9). The removal of sequences representing simple repetitions was applied using

PRINSEQ (66) at a DUST threshold of 7 (-lc_method dust, -lc_threshold 7). Contamination of the viral metagenomes with sequences of cellular origin was determined by the presence of 16S rRNA gene sequences using the SortmeRNA software (67). Details for the number of obtained sequences are shown in Table S1.

**Viral metagenome assembly and gene prediction.** Viral metagenomes were assembled using de Bruijn graphs as implemented by the SPAdes assembler (68) in metagenomic mode (metaSPAdes). Assembled sequences from each metagenome were *in silico* decontaminated using BLASTN (69) (-evalue 0.00001, query coverage of ≥5%) against *Bacteria*, *Archaea*, *Eukarya*, and phiX-174 sequences in the NCBI Nucleotide database. Contigs aligned to cellular genomes were used to search for temperate viruses using PHASTER (70). Only "intact" regions under the PHASTER completeness classification were returned to the viral contig data sets; otherwise, they were discarded. Contaminated contigs were used to remove contaminated reads, the results of which are hereafter referred to as clean reads, using Bowtie2 (71) (-end-to-end -very sensitive -N 1 –un-conc). To test the accuracy of the comparative abundance analyses and the coverage of each viral metagenomic data set, clean reads were analyzed using Nonpareil software, which examines the redundancy between individual reads using k-mers (72). Finally, Prodigal software (73) was used to predict protein-coding regions (-p meta -n).

**Viral metagenome read comparison and taxonomic assignment.** The pairwise mutation distance from the samples was calculated using the clean reads and the MinHash dimensionality-reduction technique implemented in MASH (74) (-k 19 and -s 9,999,999). The genetic distance between samples (MASH distance) was visualized by hierarchical clustering (hclust function in R) using the minimal increase of the sum of squares method (Ward.d2 function in R). Clean reads were aligned against viral sequences (taxonomy ID 10239) in the NCBI Nucleotide database using DIAMOND (75) (-evalue 0.0000001 -sensitive). Mapped sequences were parsed using the lowest common ancestor algorithm and normalized to the smallest sample size in MEGAN 6 (76) (lowest common ancestor [LCA] score of 50).

**Quantitative analyses: viral protein clusters and viral OTUs.** Predicted proteins of ≥60 amino acids (aa) were used to form protein clusters (PCs) as described in references 77 and 78. Proteins were clustered by Cd-hit (79) at 60% identity and 80% coverage (77, 78). After clustering, all PCs were quantified through read mapping in each viral metagenome using Bowtie2 (71) (-end-to-end -very sensitive -N 1). PC relative abundances were normalized by gene length and library size of each viral metagenome as described in references 77 and 78). The PCs were functionally annotated using the representative sequence of each PC against the Pfam database (80) using hmmscan options (–cut_ga) implemented on HMMER3 (81). Finally, the normalized PC abundances were used to calculate alpha diversity via the Shannon H index, evenness using Pielou's index, and species richness using the expected species richness with the vegan package in R (82). Beta diversity using Bray-Curtis distance was also calculated with the vegan package in R (82).

Viral contigs of ≥5 kb were used to generate viral operational taxonomic units (vOTUs), defined here as contigs from all samples clustered at ≥95% identity and ≥80% coverage using the nucmer algorithm implemented in MUMmer3 (83) as described in references 43, 84, 85. The resulting vOTUs data set was screened to identify prophage sequences using PHASTER designating as lysogenic only those vOTUs defined as intact regions. Additionally, sequences from temperate viruses, defined as intact regions in the PHASTER analysis of Porcelana MAGs (see next section), were added to this vOTUs set.

Each vOTU was quantified in each viral metagenome through read mapping using Bowtie2 (71) (-end-to-end -very sensitive -N 1), and the resulting SAM file was parsed by the BBmap pileup script (Bushnell B. - https://sourceforge.net/projects/bbmap/). Only when reads covered ≥75% of the vOTU sequence length at ≥95% identity was the vOTU considered present in the sample. Relative abundances of viral vOTUs were normalized using the trimmed mean of M-values algorithm, implemented in the edgeR package (86), and vOTU sequence length as described in reference 43. The normalized abundance was used to calculate species alpha and beta diversity with the vegan package in R (82). Alpha diversity was measured via the Shannon H index, evenness using Pielou's index, and species richness using the expected species richness with the vegan package in R (82). Beta diversity was measured through Bray-Curtis distance using the vegan package in R (82).

The most abundant vOTUs and the lysogenic vOTUs were annotated using the VIBRANT (87) software (option -virome). Gene maps were constructed in R using the gggenes package.

**Identification of prophages and CRISPR spacers in Porcelana metagenome-assembled genomes.** Assemblies of published Porcelana microbial mat metagenomes (21, 29) were taxonomically grouped into MAGs as described in references 21 and 31, with modifications to the quality selection and taxonomic classification steps. Briefly, metagenomes were assembled using SPAdes assembler (metaSPAdes) (68). Assemblies were taxonomically grouped (binned) using the expectation-maximization (EM) algorithm implemented in MaxBin 2.0 (88). The completeness and contamination of each bin were assessed using CheckM (89), and taxonomic classification was performed using GTDB taxonomy (90) with selection criterion of quality score of ≥50 (≥90% complete, ≤10% contaminated) (39). A total of 34 high-quality MAGs from 9 phyla were recovered and analyzed for the presence of temperate viruses using PHASTER (70). Only intact regions under the completeness classification were subsequently analyzed.

CRISPR loci were identified in the 34 MAGs obtained from Porcelana using the CRISPRFinder tool (91). Spacers from CRISPR loci were mapped to the vOTU set using Bowtie2 (71) (-end-to-end -very sensitive -N 1).

**Statistical analysis.** Bray-Curtis dissimilarity matrices (PCs and vOTUs) were visualized with a principal coordinates analysis (PCoA) plot using the ampvis2 R package (92). The significance of the ordination was assessed by 999 permutations. Vectors of significant hot spring physicochemical factors (pH and temperature) ($P < 0.05$) were fitted onto the PCoA ordination space using the envfit function of the

vegan R package (82) with 999 random permutations. Rank-based permutation tests of Brunner-Dette-Munk (BDM test, "BDM.2way" function in the asbio R package) were used to analyze the influence of MitC induction and sampling site in the PC and vOTU abundance data. Later, a pairwise Wilcoxon test (function pairwise.wilcox.test in R) with Bonferroni correction was used to determine which group differences were statistically significant. Venn diagrams were used to analyze all possible PC and vOTU relationships between samples using Venny (93) and were plotted in R (draw.quad.venn function in VennDiagram R package). The differential abundance (based on read mapping) of vOTUs (false-discovery rate [FDR] $\leq$ 0.01) and PCs (FDR $\leq$ 0.05) between MitC-induced and natural samples was assessed using a paired test in edgeR (86), phyloseq_to_edgeR function implemented in the phyloseq package (94, 95).

**Monopartite networks.** A protein monopartite network implemented in the vContact2 tool of the iVirus platform (54, 55) was constructed using the vOTU set (considered here as viral genomes) from all samples. Briefly, predicted proteins from the vOTUs were compared by DIAMOND (75) in an all-versus-all pairwise comparison (-evalue 0.00001, bitscore 50). Protein clusters were subsequently identified using the ClusterONE algorithm (96) based on DIAMOND E values with an inflation value of 2, building protein cluster profiles for each genome and generating a similarity network. For network visualization, we used an edge-weighted spring-embedded model implemented in Cytoscape 3.7.1 (97), which places the genomes that share more proteins closer to each other, forming viral clusters. Viral clusters (modules) were organized according to their predicted host from our CRISPR spacer analysis and clustered to reference genomes with ICTV taxonomy on viral RefSeq release 85.

**Data availability.** Raw sequences are publicly available under NCBI SRA BioProject PRJNA690782. Assembled vOTUs sequences and supplemental material are publicly available at GitHub https://github.com/phageattack/Porcelana-viromes.

## SUPPLEMENTAL MATERIAL

Supplemental material is available online only.
**SUPPLEMENTAL FILE 1**, PDF file, 2.1 MB.

## ACKNOWLEDGMENTS

We want to thank Christina Ridley for her help in the language editing of the manuscript and for her valuable opinion on it. We are also grateful to Huinay Scientific Field Station for making our work in Porcelana hot spring possible.

S.G.L. and B.D. conceived and designed the experiments. S.G.L. performed the experiments. S.G.L., F.S., O.S., and B.D. analyzed the data. S.G.L., F.S., and B.D. wrote the paper. S.G.L., F.S., C.R., L.Q., and B.D. reviewed and produced the manuscript's final version.

This work was financially supported by Ph.D. scholarships ANID N°21130667 and N° 21172022, ANID-FONDECYT grants N°1150171 and N°1190998, ANID-ECOS160025, and Iniciativa de Investigación UnACh 2020-132-Unach.

We declare that the research was conducted in the absence of any commercial or financial relationships that could be construed as a potential conflict of interest.

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
