## [Reviewer comments · Microbiology Spectrum]

Microbiology Spectrum

Unveiling ecological and genetic novelty within lytic and lysogenic viral communities of hot spring phototrophic microbial mats.

Sergio Guajardo-Leiva, Fernando Santos, Oscar Salgado, Christophe Regeard, Laurent Quillet, and Beatriz Díez

Corresponding Author(s): Beatriz Díez, Pontifical Catholic University of Chile

Review Timeline:

Submission Date:	June 27, 2021
Editorial Decision:	July 24, 2021
Revision Received:	August 25, 2021
Editorial Decision:	October 15, 2021
Revision Received:	October 20, 2021
Accepted:	October 21, 2021

Editor: Wei-Hua Chen

Reviewer(s): Disclosure of reviewer identity is with reference to reviewer comments included in decision letter(s). The following individuals involved in review of your submission have agreed to reveal their identity: Leonardo Erijman (Reviewer #3)

Transaction Report:

DOI: <https://doi.org/10.1128/Spectrum.00694-21>

July 24, 2021

Prof. Beatriz Díez
Pontifical Catholic University of Chile
Department of Molecular Genetics and Microbiology
Santiago
Chile

Re: Spectrum00694-21 (Unveiling novelty within lytic and lysogenic viral communities of hot spring phototrophic microbial mats.)

Dear Prof. Beatriz Díez:

Thank you for submitting your manuscript to Microbiology Spectrum. I am happy to invite you to submit a revision of your manuscript. The reviewers' comments are generally positive. However, there are still some major concerns, especially the ones mentioned by reviewer #2. So please revise your manuscript accordingly, and provide point-to-point responses to the comments.

I notice that a reviewer requested you to cite several papers. You can ignore the request if the said publications are irrelevant. In any case, you need provide a brief summary of the papers and explain why they are relevant or irrelevant to your study.

When submitting the revised version of your paper, please provide (1) point-by-point responses to the issues raised by the reviewers as file type "Response to Reviewers," not in your cover letter, and (2) a PDF file that indicates the changes from the original submission (by highlighting or underlining the changes) as file type "Marked Up Manuscript - For Review Only". Please use this link to submit your revised manuscript - we strongly recommend that you submit your paper within the next 60 days or reach out to me. Detailed information on submitting your revised paper are below.

Link Not Available

Sincerely,

Wei-Hua Chen

Journals Department
Reviewer comments:

Reviewer #1 (Comments for the Author):

The manuscript entitled " Unveiling novelty within lytic and lysogenic viral communities of hot spring phototrophic microbial mats " tries to investigate the mechanisms by which lysogenic viruses influence their host-microbial community.

The authors examined the compositional changes of viral communities at two different sites at the genomic and gene levels following in-situ mitomycin C induction experiments in PMMs. The presence of integrated prophage sequences in environmental metagenome-assembled genomes from published Porcelana PMM metagenomes was also examined.

They argued that their results showed a nexus between the ecological role of the host (metabolism) and the type of viral lifestyle in thermophilic PMMs. The most abundant lytic viral groups corresponded to cyanophages, while lysogenic viruses were exclusively related to chemoheterotrophic bacteria from the phyla Proteobacteria, Firmicutes and Actinobacteria.

This article presented important information about dynamics of viral communities in hot springs virome sequencing and metagenomic analysis. The sample size in the experiments in this study was only four, but these efforts contribute to a better understanding of viral ecology especially in extreme environments like hot springs. This article has a lot of merits especially considering the knowledge gap in understanding viral ecology.

However, when presenting the background and discussing the results, the manuscript lacks efforts in citing more related references. It is important to keep track of the newest development in this field and keep references updated. Overall, I think the experiments are valuable and should be of interest to the broad readers of Microbiology Spectrum. Please see my comments as following.

Lines 87-88: " Viruses contribute to the fitness and evolution of their host via their infective processes, where the specific viral-lifestyle has different effects on the host's ecology (1, 3).

Comment: More references should be added here. Please consider:

Liang, X., Wang, Y., Zhang, Y., Li, B. and Radosevich, M., 2021. Bacteriophage-host depth distribution patterns in soil are maintained after nutrient stimulation in vitro. *Science of The Total Environment*, 787, p.147589.

Line 93-95: "This cycle does not produce viral particles immediately after infection; however, it can switch to a productive cycle depending on multiple factors, such as virus or host genetics, virus-host ratio, host physiological state, and environmental conditions (4-6)."

Comment: New references are needed here.

For example:

Liang, X., Wagner, R.E., Li, B., Zhang, N. and Radosevich, M., 2020. Quorum sensing signals alter in vitro soil virus abundance and bacterial community composition. *Frontiers in Microbiology*, 11, p.1287.

Line 101-103: "Lytic and lysogenic dynamics have mostly been studied in aquatic environments (marine and freshwater), while a few studies have focused on sediments and soils (1, 6)."

Comment: More related references are needed.

& Line 536-537: "In other environments, lysogeny has been estimated by quantifying the viral progeny after prophage excision using the DNA damaging agent MitC (1, 6, 9)."

Roy, K., Ghosh, D., DeBruyn, J.M., Dasgupta, T., Wommack, K.E., Liang, X., Wagner, R.E. and Radosevich, M., 2020. Temporal dynamics of soil virus and bacterial populations in agricultural and early plant successional soils. *Frontiers in Microbiology*, 11, p.1494.

Liang, X., Zhang, Y., Wommack, K.E., Wilhelm, S.W., DeBruyn, J.M., Sherfy, A.C., Zhuang, J. and Radosevich, M., 2020. Lysogenic reproductive strategies of viral communities vary with soil depth and are correlated with bacterial diversity. *Soil Biology and Biochemistry*, 144, p.107767.

Brum, J.R., Hurwitz, B.L., Schofield, O., Ducklow, H.W. and Sullivan, M.B., 2016. Seasonal time bombs: dominant temperate viruses affect Southern Ocean microbial dynamics. *The ISME journal*, 10(2), pp.437-449.

Line 344-345: "Showing that it is possible to detect differentially abundant sequences between these metagenomes accurately (41)."

Comment: This sentence seems wired. Does it belong to the sentence above?

Line 543-546: " Together, our results suggest that the lytic lifestyle is dominant in viral populations that infect the most active and abundant primary producers in these mats, such as Cyanobacteria and Chloroflexi. Meanwhile, the lysogenic lifestyle is common for PMM viral populations associated with chemoheterotrophic members of the phyla Proteobacteria, Firmicutes and Actinobacteria. "

Comment: I would suggest you use past tense when describing the results, because the results only represent the situation when you did the experiment. It is hard to draw conclusions that apply to all similar environments when you only had four samples.

Reviewer #2 (Public repository details (Required)):

Viral metagenomes.

Reviewer #2 (Comments for the Author):

SUMMARY

In the manuscript "Unveiling novelty within lytic and lysogenic viral communities of hot spring phototrophic microbial mats," Guajardo-Leiva et al. obtained and investigated the genomic content of viral communities of microbial mats from two sites in Porcelana hot spring (Chile). They obtained the resident viral community as well as the viral community enriched on prophages inducible via mitomycin C. They compared their results with metagenomically-assembled genomes from published metagenomes from the same hot spring. Their main result is that abundant viruses infecting the main primary producer are lytic, while lysogeny is more prevalent among chemoheterotrophic bacteria. They interpret the result ecologically associating lysis to the primary producers in oligotrophic conditions ("kill-the-winner") and lysogeny in the heterotrophic community in eutrophic conditions ("piggy-back-the-winner"). Independently on the analysis and interpretation of the results, the technical approach to obtain the viromes was adequate and will be beneficial for the community. My major criticisms are associated with the experimental design, the level of uncertainty of the lytic and lysogenic lifestyles, and the inaccurate interpretation of some results, as detailed below. My recommendation is that the authors revise these outputs claims and be clearer on the limitations of the presented work, so the scientific community can build upon this research.

MAJOR COMMENTS

The main conclusion of the article revolves around the ecology of lytic and lysogenic life cycles in hot springs mats. However, the criteria to identify lytic and temperate viruses seems incomplete and could affect the main conclusion significantly. The specific aspects discussed in points 1 to 3 focus on this key issue.

1. vOTUS were considered temperate if they aligned with cell genome fragments identified as prophages using PHASTER (lines 216-217). This is a conservative approach but potentially biased. Many temperate viruses in the community may not be identified as lysogenic because the presence of prophages and mobile elements in different parts of a genome can prevent the formation of MAGs. In other words, the complete or partial prophages identified in MAGs are those that are present in multiple hosts at a consistent genome region. This is a limitation that needs to be noted and taken into consideration.

2. Additionally, eight out of the fourteen abundant viral genomes listed in Figure 5 contain lysogenic markers. But some of these contigs were assigned a lytic lifestyle, for example, vO2 (Table 3). It is, thus, unclear if these and other contigs are actually lytic or lysogenic. The use of lysogenic markers in viral contigs to identify temperate viruses has a level of uncertainty, but it should be at least noted that some contigs could be potentially lysogenic based on these markers. There is also a significant amount of not assigned lifestyles (as noted in Table 3 but not discussed in the text). The authors used VIBRANT for annotation, and it would be informative to provide the lytic and lysogenic classification of VIBRANT on these and the other contigs. Overall, I would recommend the authors to be more thorough in assigning degrees of certainty to the lytic and lysogenic lifestyles identified.

3. Temperate viruses can also be identified by mapping the enriched contigs (induced prophages) from the MitC viromes. This mapping analysis, however, was either missing in the manuscript or was not obvious. It is not clear if there was a technical reason that prevented such identification. This should be clarified.

4. It is claimed that "the viral communities from the same site were more similar to each other irrespective of their natural or induced condition" (lines 399-401). This observation is repeated in other parts. However, Figure 2 indicates that this observation holds for the P55 site, but it does not seem accurate for the P50 site. The distance between P50Nat and P50MitC is similar to the distance with respect to the P55 site. The authors should align the claims with the actual results and modify the discussion accordingly.

5. The taxonomic identification is based on mapping to NCBI. Given the typical diversity of viromes compared to public databases, I would expect that many contigs may not have a significant match. The same applies to the identification of hosts. This is a common situation in viromic analyses. But the results and discussion section do not seem to take notice of this frequent result. It should be made clearer what is the percentage of contigs that were not possible to assign a taxon or a host. Depending on this percentage, it should be highlighted that the interpretation and conclusion of the results may be sensitive to these unknowns.

MINOR COMMENTS

1. It is unclear what the term "unveiling novelty" in the title is referring to. I would recommend revising the title.
2. The article did not mention the rationale behind choosing the two sites. Was this a pragmatic decision? Was the difference in temperatures and pH between sites a coincidence, or was that intended? Is there any other feature between the sites that would be worth noting for those unfamiliar with the Porcelana hot spring?
3. Line 290: I recommend using the term "dissimilarity matrices" instead of "distance matrices." Strictly speaking, the Bray-Curtis dissimilarity is not a well-defined distance mathematically. It does not satisfy the triangle inequality.
4. Line 1026 (references): "asdetermined" should be "as determined."
5. In Figure 1, I would recommend using the same order of site-treatment in panels a) and b) to facilitate the interpretation of the results. Panel b) should include the gradient color scale used.

Reviewer #3 (Public repository details (Required)):

The authors sequenced viral metagenomes of samples from phototrophic microbial mats growing at two different temperatures

Reviewer #3 (Comments for the Author):

This is my first evaluation of the manuscript, which used viral metagenomics and genome-resolved metagenomics of viral hosts to analyze changes of viral communities in a thermophilic phototrophic microbial mat after mitomycin C induction. The authors concluded that viral dynamics are consistent with the "kill the winner" or the "piggyback the winner" models, depending on their host lifestyles.

I have in part guided my review on the reply to reviewers from a previous submission. Although the authors have addressed all the issues raised by the reviewers, there are limitations in the study that should temper the conclusions. However, I think that the study has merit and the debatable points are presented with appropriate caution. Therefore, I believe that the work may be a useful contribution to the literature and could be the basis for future work. I also believe that the datasets will serve as a useful resource for the phage research community

Staff Comments:

Preparing Revision Guidelines

For complete guidelines on revision requirements, please see the Instructions to Authors at [link to page]. **Submissions of a paper that does not conform to Microbiology Spectrum guidelines will delay acceptance of your manuscript.**

Please return the manuscript within 60 days; if you cannot complete the modification within this time period, please contact me. If you do not wish to modify the manuscript and prefer to submit it to another journal, please notify me of your decision immediately so that the manuscript may be formally withdrawn from consideration by Microbiology Spectrum.

If you would like to submit an image for consideration as the Featured Image for an issue, please contact Spectrum staff.

If your manuscript is accepted for publication, you will be contacted separately about payment when the proofs are issued;

please follow the instructions in that e-mail. Arrangements for payment must be made before your article is published. For a complete list of **Publication Fees**, including supplemental material costs, please visit our website.

Reviewer comments:

Reviewer #1

Please see my comments as following.

R: Lines 87-88: "Viruses contribute to the fitness and evolution of their host via their infective processes, where the specific viral-lifestyle has different effects on the host's ecology (1, 3).

Comment: More references should be added here. Please consider:

Liang, X., Wang, Y., Zhang, Y., Li, B. and Radosevich, M., 2021. Bacteriophage-host depth distribution patterns in soil are maintained after nutrient stimulation in vitro. *Science of The Total Environment*, 787, p.147589.

A: The work suggested by the reviewer is quite interesting; however, the main topic of Liang et al., 2021 is the effect of nutrient stimulation over the bacterial and viral community dynamics and virus-host interactions in different soil types (Mollisol and Alfisol) and horizons. The specific work of Liang et al., 2021 does not provide new evidence here, as it suggests the same mechanisms that are already treated in the existing references. For example, Liang et al., 2021 mention the following mechanisms in which the viral-lifestyle can influence the ecology of their host: "lytic infections can control population abundances and community diversity via lysis and impact biogeochemical cycles via the viral shunt, while lysogenic conversions may also affect the fitness of hosts (e.g., by introduction of auxiliary metabolic genes and immunity to new invading viruses).", which also appear in the cited references.

R: Line 93-95: "This cycle does not produce viral particles immediately after infection; however, it can switch to a productive cycle depending on multiple factors, such as virus or host genetics, virus-host ratio, host physiological state, and environmental conditions (4-6)."

Comment: New references are needed here.

For example:

Liang, X., Wagner, R.E., Li, B., Zhang, N. and Radosevich, M., 2020. Quorum sensing signals alter in vitro soil virus abundance and bacterial community composition. *Frontiers in Microbiology*, 11, p.1287.

A: The main topic of Liang et al., 2020 is the activation of the lysogeny-lysis switch of some temperate phages by quorum sensing (QS) systems. They used eight (different sizes and structure) acyl-homoserine lactones (AHLs) as QS signaling molecules to evaluate the phage responses of lysogeny-lysis switching in mixed bacterial communities from a single soil source in-vitro. They demonstrate that a broad range of bacterial taxonomic groups was susceptible to prophage induction by AHLs and that transitions from lysogeny to lysis of temperate phages have pivotal roles in influencing bacterial community structure. Therefore, the publication suggested by the reviewer is very relevant because it adds a new factor (the quorum sensing) that we had not considered previously in the activation of the lysogenic-lytic switch and has been added to the references.

R: Line 101-103: "Lytic and lysogenic dynamics have mostly been studied in aquatic environments (marine and freshwater), while a few studies have focused on sediments and soils (1, 6)."

Comment: More related references are needed.

A: Both publications made a compressive review of the available bibliography until 2017. However, we agree with the reviewer about the necessity of include more actualized references. We have added some of the proposed by the reviewer and the sentences "and references therein."

R: Line 536-537: "In other environments, lysogeny has been estimated by quantifying the viral progeny after prophage excision using the DNA damaging agent MitC (1, 6, 9)."

Roy, K., Ghosh, D., DeBruyn, J.M., Dasgupta, T., Wommack, K.E., Liang, X., Wagner, R.E. and Radosevich, M., 2020. Temporal dynamics of soil virus and bacterial populations in agricultural and early plant successional soils. *Frontiers in Microbiology*, 11, p.1494.

A: The main objective of Roy et al., 2020 was to assess the spatial and temporal variability in viral abundance, community structure, and the frequency of lysogeny among soil microbial communities and to correlate those properties with soil edaphic factors in agricultural, early plant successional, and mid-successional forest soils. The main results indicated that the structure of soil viral communities was influenced by land use and season and that the free extracellular viruses and the inducible prophages have different responses to the environmental factors. Interestingly, the work mention that the high heterogeneity of soil physical properties (even at small distances) makes the study of environmental viral ecology at the field scale challenges.

We agree with the reviewer, because the article complements the existing references and it updates the listed references, including agricultural and forest soils, to environments where mitomycin C has been used to study the induction of prophages.

Liang, X., Zhang, Y., Wommack, K.E., Wilhelm, S.W., DeBruyn, J.M., Sherfy, A.C., Zhuang, J. and Radosevich, M., 2020. Lysogenic reproductive strategies of viral communities vary with soil depth and are correlated with bacterial diversity. *Soil Biology and Biochemistry*, 144, p.107767.

A: The main objective of Liang et al., 2020 was to determine the dependence of viral abundance and reproductive strategy with soil depth in two soils formed from contrasting parent materials (eolian sand-glacial outwash vs. loess plain). To accomplish this, they investigated the viral and microbial abundances, virus-to-bacteria ratio, lysogenic fractions among microbial communities, and correlation of viral abundances with bacterial communities in two vertical soil profiles. Their results showed that the prevalence of lysogeny was positively correlated with soil depth and that the bacterial community diversity was positively correlated with virus abundance in soil. Moreover, they showed that shifts in bacterial taxonomic composition coincided with the differences in the viral abundances and reproductive strategies in soils.

We agree with the reviewer, since the article updates the listed references and includes the vertical soil profiles to environments where mitomycin C has been used to study the induction of prophages.

Brum, J.R., Hurwitz, B.L., Schofield, O., Ducklow, H.W. and Sullivan, M.B., 2016. Seasonal time bombs: dominant temperate viruses affect Southern Ocean microbial dynamics. *The ISME journal*, 10(2), pp.437-449.

A: The publication from Brum et al., 2016 targeted the WAP region of the Southern Ocean to evaluate if the bacterial activity determines the viral replication strategies. Additionally, they assess the contribution of temperate viruses to the viral communities in polar aquatic regions. For these objectives, they tracked the temporal changes in lysogeny and lytic viral replication about bacterial production and abundance and compared the WAP dsDNA viral communities with those from lower-latitude marine systems. Their results demonstrated that temperate viruses dominate the WAP dsDNA viral communities, utilizing lysogeny when bacterial production is low and switching to lytic replication when bacterial production increases.

The work by Brum et al., 2015 is already cited in the work of reference 6 (Knowles et al., 2017), so we have added the sentence "and references therein."

R: Line 344-345: "Showing that it is possible to detect differentially abundant sequences between these metagenomes accurately (41)."

Comment: This sentence seems wired. Does it belong to the sentence above?

A: We totally agree with the reviewer since the sentence was part of the previous sentence. We have modified the text as follows "The results showed coverages in the range of 0.86 to 0.92 above the recommended coverage threshold of 0.6 and with coverage differences less than twofold between datasets, confirming that it is possible to detect differentially abundant sequences between these metagenomes accurately."

R: Line 543-546: "Together, our results suggest that the lytic lifestyle is dominant in viral populations that infect the most active and abundant primary producers in these mats, such as Cyanobacteria and Chloroflexi. Meanwhile, the lysogenic lifestyle is common for PMM viral populations associated with chemoheterotrophic members of the phyla Proteobacteria, Firmicutes and Actinobacteria. "

Comment: I would suggest you use past tense when describing the results, because the results only represent the situation when you did the experiment. It is hard to draw conclusions that apply to all similar environments when you only had four samples.

A: We totally agree with the reviewer, and we have changed the whole sentence to past tense.

"Together, our results suggested that the lytic lifestyle was dominant in viral populations that infected the most active and abundant primary producers in these mats, such as Cyanobacteria and Chloroflexi. Meanwhile, the lysogenic lifestyle was common for PMM viral populations associated with chemoheterotrophic members of the phyla Proteobacteria, Firmicutes, and Actinobacteria."

Reviewer #2 (Public repository details (Required)):

Viral metagenomes.

A: Viral metagenomes are publicly available at NCBI SRA BioProject PRJNA690782.

This information is listed in lines 314-317 in the Data availability section of MM.

"Data availability

Raw sequences are publicly available under NCBI SRA BioProject PRJNA690782. Assembled vOTUs sequences and supplementary material are publicly available at GitHub <https://github.com/phageattack/Porcelana-viromes>."

Reviewer #2 (Comments for the Author):

MAJOR COMMENTS

R: The main conclusion of the article revolves around the ecology of lytic and lysogenic life cycles in hot springs mats. However, the criteria to identify lytic and temperate viruses seems incomplete and could affect the main conclusion significantly. The specific aspects discussed in points 1 to 3 focus on this key issue.

1. vOTUS were considered temperate if they aligned with cell genome fragments identified as prophages using PHASTER (lines 216-217). This is a conservative approach but

potentially biased. Many temperate viruses in the community may not be identified as lysogenic because the presence of prophages and mobile elements in different parts of a genome can prevent the formation of MAGs. In other words, the complete or partial prophages identified in MAGs are those that are present in multiple hosts at a consistent genome region. This is a limitation that needs to be noted and taken into consideration.

A: We have used three different strategies to determine the presence of putative lysogenic viruses. The first was to search for prophage sequences in the vOTUs dataset using lysogenic markers implemented on PHASTER. The second one was to search for prophage sequences in Porcelana MAGs also with PHASTER. Finally, the third was to search for vOTUs statistically more abundant in the MitC induced samples than in the natural ones using a paired test implemented in the function `Phyloseq_to_EdgeR`.

We realized that the first strategy was unclear in the MM section, so we included the following sentence in lines 248-250. "The resulting vOTUs dataset was screened to identify lysogenic viral sequences using PHASTER designating as lysogenic only that vOTUs defined as "intact" regions."

We also agree with the reviewer that we have used a conservative approach by selecting PHASTER. However, in an environment where viral communities are unknown, we opted to reduce the number of possible false positives in identifying lysogenic viruses from the vOTUs dataset. Likewise, the use of MAGs to search for integrated prophages, we chose this strategy as it was the best way to link a prophage to a known host securely. Another option was to directly use the contigs assembled from the cellular metagenomes, but identifying the possible host would have been difficult or impossible. According to the above, we again prefer to be more conservative and better fulfill our objective of link viruses, their lifestyles, and their hosts.

Despite the above, we decided to search for possible lysogenic genomes in the vOTUs dataset using differential abundance analyses (`Phyloseq_to_EdgeR`) that have proven useful in searching for biomarker taxa of interest in analyses based on amplicon sequencing (mostly 16S rRNA).

To make these limitations more apparent to the reader, we have added the following sentence in the Discussion section, lines 632-634. "Another possible explanation for this low number of prophage sequences found in Porcelana MAGs is the inherent difficulty in obtaining mobile elements in MAGs generated from short reads."

2. Additionally, eight out of the fourteen abundant viral genomes listed in Figure 5 contain lysogenic markers. But some of these contigs were assigned a lytic lifestyle, for example, vO2 (Table 3). It is, thus, unclear if these and other contigs are actually lytic or lysogenic. The use of lysogenic markers in viral contigs to identify temperate viruses has a level of uncertainty, but it should be at least noted that some contigs could be potentially lysogenic based on these markers. There is also a significant amount of not assigned lifestyles (as noted in Table 3 but not discussed in the text). The authors used VIBRANT for annotation, and it would be informative to provide the lytic and lysogenic classification of VIBRANT on these and the other contigs. Overall, I would recommend the authors to be more thorough in assigning degrees of certainty to the lytic and lysogenic lifestyles identified.

A: We have extensively discussed the detail of the vOTUs shown in Table 3 that have information regarding putative hosts or that were part of clusters with reference viruses (lines 649-682). Likewise, we have discussed the incongruities (if they exist) between the lifestyle of the reference genomes and the presence of marker genes or higher abundance in MitC samples, indicating the need for further studies to determine the authentic lifestyle of those viruses.

To make this more apparent to readers, we have added the vOTUs numbers in the discussion section of the revised manuscript and the following sentence "Finally, there were eight (vO5,6,7,10,14,16,17 and vO22) of the 23 most abundant vOTUs for which it was not possible to establish a lifestyle based on references. Two of these eight vOTUs (vO6 and vO22) showed the presence of recombinases, which could suggest a lysogenic lifestyle." in lines 690-693 to cover the viruses without identified lifestyle in Table 3.

Finally, we did not use Vibrant classification since there is no certainty about Lytic/Lysogenic classification. Briefly, Vibrant classify viruses as lysogenic if they are excised from a larger (putative host) contig or if an integrase is identified. In this way, all contigs not considered lysogenic are classified as lytic, but they can contain lysogenic viruses that VIBRANT could not distinguish. <https://github.com/AnantharamanLab/VIBRANT/issues/16>

3. Temperate viruses can also be identified by mapping the enriched contigs (induced prophages) from the MitC viromes. This mapping analysis, however, was either missing in the manuscript or was not obvious. It is not clear if there was a technical reason that prevented such identification. This should be clarified.

A: Yes, we have used read mapping to identify possible lysogenic contigs. However, we have specifically used differential abundance analyzes (Phyloseq_to_EdgeR) from count tables based on read mapping. This analysis only reported vOTUs statistically more abundant in the MitC samples than in the natural ones. However, to make this more evident to the reader, we have added the following sentences "through read mapping" (lines 238 and 254) and "based on read mapping" (lines 298-299) to the MM section in the revised version of this manuscript.

4. It is claimed that "the viral communities from the same site were more similar to each other irrespective of their natural or induced condition" (lines 399-401). This observation is repeated in other parts. However, Figure 2 indicates that this observation holds for the P55 site, but it does not seem accurate for the P50 site. The distance between P50Nat and P50MitC is similar to the distance with respect to the P55 site. The authors should align the claims with the actual results and modify the discussion accordingly.

A: Effectively, we claimed that "Altogether, these results show that the viral communities from the same site were more similar to each other irrespective of their natural or induced condition and that MitC induction generates shifts in the viral communities that account for ~30% of the community structure variance." based on the evidence that PCoA shows that differences between sites explain the 55-60% of the variance between samples. However, we think that the sentence is poorly formulated and can lead to erroneous conclusions, so we have modified the text to the following: "Altogether, these results show that dissimilarities between sites contribute to a greater extent (55-60%) to the community structure variance compared to MitC induction that represents the ~30% of the variance."

5. The taxonomic identification is based on mapping to NCBI. Given the typical diversity of viromes compared to public databases, I would expect that many contigs may not have a significant match. The same applies to the identification of hosts. This is a common situation in viromic analyses. But the results and discussion section do not seem to take notice of this frequent result. It should be made clearer what is the percentage of contigs that were not possible to assign a taxon or a host. Depending on this percentage, it should be highlighted that the interpretation and conclusion of the results may be sensitive to these unknowns.

A: The taxonomic assignment was performed by mapping the clean reads (not the contigs) against the NCBI NR database using DIAMOND (blastx mode) and parsing the result using the last common ancestor algorithm in MEGAN. The detail of this methodology is described in the MM section (lines 228-231). Likewise, this detail is provided in the legend of Figure 1B "Metagenomic reads were taxonomically classified by the LCA algorithm through local alignment to NCBI nr database."

For the contigs, the taxonomic classification was performed based on the results of the protein network (analysis by vContact2) for the contigs with relative abundances > 1% and for putatively lysogenic contigs (vOTUs identified by the differential abundance analyzes and/or PHASTER) that appear in Table 2 and Table 3.

We agree with the reviewer that mapping reads against general databases such as NCBI NR returned many unknowns. The percentage (5-11%) of mapped reads against the NCBI NR database is available in Supplementary Table S1. However, following the recommendations of the reviewer, we have added the following sentence in the Results section (lines 341-342) "Despite the low number of mapped reads (in the range of 155000 to 937000)". This is complementary to the sentence that was already in the discussion (lines 554-556) "Furthermore, to avoid bias due to the low number of sequences with taxonomic annotations, we cataloged viral sequences using a database-independent approach by quantifying PCs and vOTUs."

MINOR COMMENTS

1. It is unclear what the term "unveiling novelty" in the title is referring to. I would recommend revising the title.

A: We agree with the reviewer and have changed the title to a more meaningful version, "Unveiling ecological and genetic novelty within lytic and lysogenic viral communities of hot spring phototrophic microbial mats."

2. The article did not mention the rationale behind choosing the two sites. Was this a pragmatic decision? Was the difference in temperatures and pH between sites a coincidence, or was that intended? Is there any other feature between the sites that would be worth noting for those unfamiliar with the Porcelana hot spring?

A: We choose both sites based on our previous work in Porcelana (Guajardo-Leiva et al., 2018; Alcorta et al., 2018 and Alcamán-Arias et al., 2018) since they provide different environments (pH and Temperature) and congruently different microbial communities in the same hot spring. Likewise, there is no other feature between the sites that would be worth noting.

3. Line 290: I recommend using the term "dissimilarity matrices" instead of "distance matrices." Strictly speaking, the Bray-Curtis dissimilarity is not a well-defined distance mathematically. It does not satisfy the triangle inequality.

A: We agree with the reviewer, and we have corrected the text in the revised version of this manuscript.

4. Line 1026 (references): "asdetermined" should be "as determined."

A: We have corrected the text in the revised version of this manuscript.

5. In Figure 1, I would recommend using the same order of site-treatment in panels a) and b) to facilitate the interpretation of the results. Panel b) should include the gradient color scale used.

A: We agree with the reviewer, and we changed the image according to that.

Reviewer #3 (Public repository details (Required)):

A: Viral metagenomes are publicly available at NCBI SRA BioProject PRJNA690782. This information is listed in lines 314-317 in the Data availability section of MM.

"Data availability

Raw sequences are publicly available under NCBI SRA BioProject PRJNA690782. Assembled vOTUs sequences and supplementary material are publicly available at GitHub <https://github.com/phageattack/Porcelana-viromes>."

The authors sequenced viral metagenomes of samples from phototrophic microbial mats growing at two different temperatures

Reviewer #3 (Comments for the Author):

R: This is my first evaluation of the manuscript, which used viral metagenomics and genome-resolved metagenomics of viral hosts to analyze changes of viral communities in a thermophilic phototrophic microbial mat after mitomycin C induction. The authors concluded that viral dynamics are consistent with the "kill the winner" or the "piggyback the winner" models, depending on their host lifestyles.

I have in part guided my review on the reply to reviewers from a previous submission. Although the authors have addressed all the issues raised by the reviewers, there are limitations in the study that should temper the conclusions. However, I think that the study has merit and the debatable points are presented with appropriate caution. Therefore, I believe that the work may be a useful contribution to the literature and could be the basis for future work. I also believe that the datasets will serve as a useful resource for the phage research community

A: We are grateful for the reviewer comments.

October 15, 2021

Prof. Beatriz Díez
Pontifical Catholic University of Chile
Department of Molecular Genetics and Microbiology
Santiago
Chile

Re: Spectrum00694-21R1 (Unveiling ecological and genetic novelty within lytic and lysogenic viral communities of hot spring phototrophic microbial mats.)

Dear Prof. Beatriz Díez:

Thank you for submitting your manuscript to Microbiology Spectrum. As you will see your paper is very close to acceptance. Please modify the manuscript along the lines reviewer #1 has recommended. As these revisions are text-only, I expect that you should be able to turn in the revised paper in less than 30 days, maybe sooner; I will then accept it without further external review.

You will find the reviewers' comments below.

When submitting the revised version of your paper, please provide (1) point-by-point responses to the issues I raised in your cover letter, and (2) a PDF file that indicates the changes from the original submission (by highlighting or underlining the changes) as file type "Marked Up Manuscript - For Review Only". Please use this link to submit your revised manuscript. Detailed information on submitting your revised paper are below.

Link Not Available

Sincerely,

Wei-Hua Chen

Reviewer comments:

Reviewer #1 (Public repository details (Required)):

This study includes metagenomic sequencing data, and the large datasets have been appropriately deposited in a public repository as described in the manuscript. The raw sequences are publicly available under NCBI SRA BioProject PRJNA690782. Assembled vOTUs sequences and supplementary material are publicly available at GitHub.

Reviewer #1 (Comments for the Author):

I thank the authors for addressing all my previous comments. I have no other major comments at this point.

Some minor suggestions are provided as following.

L40-43: "One of the most abundant lytic viral groups corresponds to cyanophages, which would infect the cyanobacteria Fischerella, the most active and dominant primary producer in thermophilic PMMs."

Suggestion: "corresponds to" was used in this sentence, but past tense was used in the following sentence in describing the other finding. I would suggest consistently using past tense to describe your results.

L48-50: "The above has direct implications in viral ecology, where the lysogenic-lytic switch is related to the abundance of nutrients, microbial density, and the types of metabolism prevailing in the host community."

Suggestion: I think this sentence, as the ending sentence of the abstract, could be improved.

L62-65: "The importance of our research is to explore the differences in the composition of natural and induced viral communities at the genome and gene level to improve our understanding of viral lifestyles in PMMs."

Suggestion: "The importance of our research is to improve our understanding of viral lifestyles in PMMs via exploring the differences in the composition of natural and induced viral communities at the genome and gene level."

L97-99: "Lytic viruses directly influence microbial community composition through predator-prey dynamics, in which the dominant or active taxa in the microbial community is selectively lysed, as described in the "kill-the-winner" ecological model "

Suggestion: "Lytic viruses directly influence microbial community composition through predator-prey dynamics, in which the dominant or active taxa in the microbial community are selectively lysed, as described in the "kill-the-winner" ecological model "

L145-152:

Suggestion: Past tense should be used in describing your results.

L342-343: "The results showed coverages in the range of 0.86 to 0.92, that are above the recommended coverage threshold of 0.6."

Comment: This sentence is not complete.

L343-345:

Suggestion: Past tense should be used in describing your results.

L400-402: "Taken together, these results show that dissimilarities between sites contribute to a greater extent (55-60%) to the variance in community structure compared to MitC induction which accounts for ~30% of the variance."

Suggestion: Past tense should be used in describing your results. "Taken together, these results show that sampling sites contributed to a greater extent (55-60%) to the variance in viral community structure than MitC induction which accounted for about 30% of the variance."

L544-548: "Together, our results suggest that the lytic lifestyle was dominant in viral populations that infected the most active and abundant primary producers in these mats, such as Cyanobacteria and Chloroflexi. Meanwhile, the lysogenic lifestyle was common among PMM viral populations with chemoheterotrophic members of the phyla Proteobacteria, Firmicutes, and Actinobacteria."

Suggestion: "Together, our results suggest that the lytic lifestyle was dominant in viral populations that infect the most active and abundant primary producers in these mats, such as Cyanobacteria and Chloroflexi, while the lysogenic lifestyle was common among PMM viral populations associated with chemoheterotrophic members of the phyla Proteobacteria, Firmicutes, and Actinobacteria."

Reviewer #2 (Public repository details (Required)):

The authors have addressed this issue. They made the viral metagenomes available in public repositories and included the information in the manuscript.

Reviewer #2 (Comments for the Author):

The authors have addressed the comments in my initial review. I don't have further comments.

Reviewer #3 (Public repository details (Required)):

The authors have already made the data available at NCBI

Reviewer #3 (Comments for the Author):

All reviewers' comments have been addressed satisfactorily in the revision. I have no further comments.

Preparing Revision Guidelines

To submit your modified manuscript, log onto the eJP submission site at <https://spectrum.msubmit.net/cgi-bin/main.plex>. Go to Author Tasks and click the appropriate manuscript title to begin the revision process. The information that you entered when you

first submitted the paper will be displayed. Please update the information as necessary. Here are a few examples of required updates that authors must address:

- point-by-point responses to the issues I raised in your cover letter
- Upload a compare copy of the manuscript (without figures) as a "Marked-Up Manuscript" file.
- Each figure must be uploaded as a separate file, and any multipanel figures must be assembled into one file.
- Manuscript: A .DOC version of the revised manuscript
- Figures: Editable, high-resolution, individual figure files are required at revision, TIFF or EPS files are preferred

Please return the manuscript within 60 days; if you cannot complete the modification within this time period, please contact me. If you do not wish to modify the manuscript and prefer to submit it to another journal, please notify me of your decision immediately so that the manuscript may be formally withdrawn from consideration by Microbiology Spectrum.

Responses to Reviewer's comments:

Reviewer #1 (Public repository details (Required)):

This study includes metagenomic sequencing data, and the large datasets have been appropriately deposited in a public repository as described in the manuscript. The raw sequences are publicly available under NCBI SRA BioProject PRJNA690782. Assembled vOTUs sequences and supplementary material are publicly available at GitHub.

Reviewer #1 (Comments for the Author):

I thank the authors for addressing all my previous comments. I have no other major comments at this point.

Some minor suggestions are provided as following.

L40-43: "One of the most abundant lytic viral groups corresponds to cyanophages, which would infect the cyanobacteria Fischerella, the most active and dominant primary producer in thermophilic PMMs."

Suggestion: "corresponds to" was used in this sentence, but past tense was used in the following sentence in describing the other finding. I would suggest consistently using past tense to describe your results.

Authors: We have changed the sentence to past tense, and the sentence is highlighted in the revised version of this manuscript.

L48-50: "The above has direct implications in viral ecology, where the lysogenic-lytic switch is related to the abundance of nutrients, microbial density, and the types of metabolism prevailing in the host community."

Suggestion: I think this sentence, as the ending sentence of the abstract, could be improved.

Authors: We have improved this sentence in the revised version of this manuscript.

L62-65: "The importance of our research is to explore the differences in the composition of natural and induced viral communities at the genome and gene level to improve our understanding of viral lifestyles in PMMs."

Suggestion: "The importance of our research is to improve our understanding of viral lifestyles in PMMs via exploring the differences in the composition of natural and induced viral communities at the genome and gene level."

Authors: We agree with the reviewer, and we have changed this sentence according to the reviewer's suggestion.

L97-99: "Lytic viruses directly influence microbial community composition through

predator-prey dynamics, in which the dominant or active taxa in the microbial community is selectively lysed, as described in the "kill-the-winner" ecological model"
Suggestion: "Lytic viruses directly influence microbial community composition through predator-prey dynamics, in which the dominant or active taxa in the microbial community are selectively lysed, as described in the "kill-the-winner" ecological model"

Authors: We agree with the reviewer, and we have changed this sentence according to the reviewer's suggestion.

L145-152:

Suggestion: Past tense should be used in describing your results.

Authors: We have revised the verb tenses and corrected the text, leaving the results in the past tense.

L342-343: "The results showed coverages in the range of 0.86 to 0.92, that are above the recommended coverage threshold of 0.6."

Comment: This sentence is not complete.

Authors: We have revised and corrected the sentence in the text.

L343-345:

Suggestion: Past tense should be used in describing your results.

Authors: We have revised the verb tenses and corrected the text, leaving the results in the past tense.

L400-402: "Taken together, these results show that dissimilarities between sites contribute to a greater extent (55-60%) to the variance in community structure compared to MitC induction which accounts for ~30% of the variance."

Suggestion: Past tense should be used in describing your results. "Taken together, these results show that sampling sites contributed to a greater extent (55-60%) to the variance in viral community structure than MitC induction which accounted for about 30% of the variance."

Authors: We agree with the reviewer, and we have changed this sentence according to the reviewer's suggestion.

L544-548: "Together, our results suggest that the lytic lifestyle was dominant in viral populations that infected the most active and abundant primary producers in these mats, such as Cyanobacteria and Chloroflexi. Meanwhile, the lysogenic lifestyle was common among PMM viral populations with chemoheterotrophic members of the phyla Proteobacteria, Firmicutes, and Actinobacteria."

Suggestion: "Together, our results suggest that the lytic lifestyle was dominant in viral populations that infect the most active and abundant primary producers in these mats,

such as Cyanobacteria and Chloroflexi, while the lysogenic lifestyle was common among PMM viral populations associated with chemoheterotrophic members of the phyla Proteobacteria, Firmicutes, and Actinobacteria."

Authors: We agree with the reviewer, and we have changed this sentence according to the reviewer's suggestion.

Reviewer #2 (Public repository details (Required)):

The authors have addressed this issue. They made the viral metagenomes available in public repositories and included the information in the manuscript.

Reviewer #2 (Comments for the Author):

The authors have addressed the comments in my initial review. I don't have further comments.

Reviewer #3 (Public repository details (Required)):

The authors have already made the data available at NCBI

Reviewer #3 (Comments for the Author):

All reviewers' comments have been addressed satisfactorily in the revision. I have no further comments.

October 21, 2021

Prof. Beatriz Díez
Pontifical Catholic University of Chile
Department of Molecular Genetics and Microbiology
Santiago
Chile

Re: Spectrum00694-21R2 (Unveiling ecological and genetic novelty within lytic and lysogenic viral communities of hot spring phototrophic microbial mats.)

Dear Prof. Beatriz Díez:

We are pleased to tell you that your manuscript has been accepted. Because it is a text only revision, this version is not sent out for external review.

We thank the reviewers for their time and energy to help the authors to improve their study, and us to screen high quality manuscripts.

I am forwarding it to the ASM Journals Department for publication. You will be notified when your proofs are ready to be viewed.

Sincerely,

Wei-Hua Chen
Editor, Microbiology Spectrum

Journals Department
Supplemental Material FOR Publication: Accept